# Multi-Parameter Dynamical Diagnostics for Upper Tropospheric and Lower Stratospheric Studies

Luis F. Millán[1], Gloria L. Manney[2,3], Harald Boenisch[4], Michaela I. Hegglin[5,6,7], Peter Hoor[8], Daniel Kunkel[8], Thierry Leblanc[9], Irina Petropavlovskikh[10], Kaley Walker[11], Krzysztof Wargan[12,13], and Andreas Zahn[4]

[1]Jet Propulsion Laboratory, California Institute of Technology, Pasadena, California, USA
[2]NorthWest Research Associates, Socorro, New Mexico, USA
[3]New Mexico Institute of Mining and Technology, Socorro, New Mexico, USA
[4]Karlsruhe Institute of Technology, Institute of Meteorology and Climate Research, Karlsruhe, Germany
[5]Institute of Energy and Climate Research, Stratosphere (IEK-7), Forschungszentrum Jülich, Jülich, Germany
[6]Department of Meteorology, University of Reading, Reading, UK
[7]Department of Atmospheric Physics, University of Wuppertal, Wuppertal, Germany
[8]Institute for Atmospheric Physics, University of Mainz, Mainz, Germany
[9]Jet Propulsion Laboratory, California Institute of Technology, Wrightwood, California, USA
[10]Cooperative Institute for Research in Environmental Sciences, National Ocean and Atmospheric Administration, Boulder, Colorado, USA
[11]Department of Physics, University of Toronto, Toronto, Canada
[12]Science Systems and Applications Inc., Lanham, Maryland, USA
[13]Global Modeling and Assimilation Office, NASA Goddard Space Flight Center, Greenbelt, USA

**Correspondence:** L. Millán (lmillan@jpl.nasa.gov)

**Abstract.** Ozone trend estimates have shown large uncertainties in the upper troposphere/lower stratosphere (UTLS) region despite multi-decadal observations available from ground-based, balloon, aircraft, and satellite platforms. These uncertainties arise from large natural variability driven by dynamics (reflected in tropopause and jet variations) as well as the strength in constituent transport and mixing. Additionally, despite all the community efforts there is still a lack of representative high-quality global UTLS measurements to capture this variability.

The Stratosphere-troposphere Processes And their Role in Climate (SPARC) Observed Composition Trends and Variability in the UTLS (OCTAV-UTLS) activity aims to reduce uncertainties in UTLS composition trend estimates by accounting for this dynamically induced variability. In this paper, we describe the production of dynamical diagnostics using meteorological information from reanalysis fields that facilitate mapping observations from several platforms into numerous geophysically-based coordinates (including tropopause and upper tropospheric jet relative coordinates). Suitable coordinates should increase the homogeneity of the air masses analyzed together, thus reducing the uncertainty caused by spatio-temporal sampling biases in the quantification of UTLS composition trends. This approach thus provides a framework for comparing measurements with diverse sampling patterns and leverages the meteorological context to derive maximum information on UTLS composition and trends and its relationships to dynamical variability.

The dynamical diagnostics presented here are the first comprehensive set describing the meteorological context for multi-decadal observations by ozonesondes, lidar, aircraft, and satellite measurements in order to study the impact of dynamical

processes on observed UTLS trends by different sensors on different platforms. Examples using these diagnostics to map multi-platform datasets into different geophysically-based coordinate systems are provided. The diagnostics presented can also be applied to analysis of greenhouse gases other than ozone that are relevant to surface climate and UTLS chemistry.

## 1   Introduction

Despite decades of space-borne, airborne, balloon-borne, and ground-based measurements, confidence in upper troposphere lower stratosphere (UTLS) long-term ozone trends remains low (e.g., Harris et al., 2015; Steinbrecht et al., 2017; Petropavlovskikh et al., 2019; Szeląg et al., 2020; Godin-Beekmann et al., 2022). For example, Ball et al. (2018), using multiple satellite measurements, reported that ozone in the lower stratosphere between 60°S and 60°N showed a statistically significant decline. Wargan et al. (2018) suggest that these downward trends were driven by enhanced isentropic transport between the tropical (20°S–20°N) and extratropical lower stratosphere while Orbe et al. (2020) suggest that in the mid-latitudes northern hemisphere these downward trends were mostly related to large-scale advection. Further, as discussed by Chipperfield et al. (2018), expanding the satellite time series from 1998-2016 to 1998-2017 changed the trend interpretation from one of decline to one of large dynamical variability, in agreement with the study by Stone et al. (2018). Ball et al. (2019) argued that while the inclusion of additional years (2017 and 2018) confounds the ozone trends in the southern hemisphere lower stratosphere, the trends in the northern hemisphere still showed a statistically significant decline. Lastly, Thompson et al. (2021), Bognar et al. (2022), and Match and Gerber (2022) showed that most of the downward trends in the tropics are attributable to increases in tropopause altitude due to tropospheric warming.

Some of the reasons for this low confidence in the UTLS trends are:

– The Brewer-Dobson circulation (BDC) and transport from the troposphere into the stratosphere and vice versa which influence the structure and composition of the UTLS (e.g., Gettelman et al., 2011) exhibit large dynamical variability, rendering statistical evaluations of ozone trends difficult and sensitive to the time period (start and end points) chosen. Many variations in UTLS ozone are associated with geographic and temporal variations in the tropopause and upper tropospheric jet streams (Pan et al., 2009; Manney et al., 2011; Schwartz et al., 2015; Albers et al., 2018; Olsen et al., 2019, and references therein).

– Long-term trends in these dynamical features (such as trends in the positions of the jets, the tropopause, and systematic changes in the BDC) confound the long-term trends in ozone due to chemistry. For example, global tropopause altitudes and the frequency of double tropopauses increased substantially between 1981 and 2015 (e.g., Xian and Homeyer, 2019); subtropical jet strength and position trends show large regional and seasonal variations, but jet altitudes have typically increased (consistent with reports of tropopause altitude increases related to climate change) while their velocities have in general increased in winter and decreased in summer between 1980 and 2014 (e.g., Manney and Hegglin, 2018); the

attribution of jet trends, particularly the polar (or "eddy-driven") jet, is an active research topic, with different studies arguing for either strengthening or weakening of the jets with climate change (e.g., Barnes and Screen, 2015; Francis, 2017; Manney and Hegglin, 2018).

- Available measurements cannot completely represent the global structure and variability of the UTLS, since the balloon-borne and ground-based measurements have limited geographical and temporal coverage; the aircraft measurements from regular passengers flights cover a limited altitude range (mostly one altitude level per flight, except during take-off and landing) while research campaigns are limited to regional coverage and short time periods; and space-borne measurements have too coarse horizontal and vertical resolutions. For example, Millán et al. (2016), using sampled and raw model fields, showed that even dense satellite sampling patterns may have up to 10% bias in the zonal mean representation of UTLS ozone.

Analyzing datasets in conventional coordinates (e.g., altitude or pressure / latitude grids using zonal means) does not account for local and regional variability impacting ozone gradients at the tropopause and the jets. It thus entangles the impact of dynamics on ozone with impacts of other processes (e.g., chemistry).To increase confidence in UTLS composition trends, the Observed Composition Trends and Variability in the UTLS (OCTAV-UTLS) Stratosphere-troposphere Processes And their Role in Climate (SPARC) activity (see www.octav-utls.net for more information) was initiated in 2018. This activity aims to analyze multi-platform measurements in coordinate systems that account for the geophysical variability of the jets and tropopause.

The use of well-suited coordinates can increase consistency in locating the measurements with respect to dynamical barriers and thus more accurately represent tracer gradients, which in turn will increase the homogeneity of the air masses mapped within the bins in such coordinate systems. For example, Pan et al. (2004), Hoor et al. (2004), and Hegglin et al. (2006) have shown that using tropopause-related coordinates (i.e., altitude relative to the tropopause) greatly reduces the scatter around the tropopause, revealing a sharp gradient between tropospheric and stratospheric air masses; Manney et al. (2011) and Olsen et al. (2019) have similarly shown that jet-relative coordinates reduce scatter in the horizontal revealing sharp gradients across the UT jets.

To derive these dynamically-based coordinates, a consistent characterization of the measurements using meteorological fields is required. In this paper we describe the generation of dynamical diagnostics for multi-platform ozone datasets. Apart from allowing mapping of the measurements into coordinates that should segregate measurements of ozone with similar characteristics, these dynamical diagnostics have already been used (among other examples) to study stratospheric and UTLS transport in several Arctic winters (e.g., Manney et al., 2009); to obtain a global view of the chemical characteristics of the extratropical tropopause transition region (Hegglin et al., 2009); to study the impact of double tropopauses on UTLS composition (Schwartz et al., 2015); to characterize biomass-burning plumes (Tereszchuk et al., 2011, 2013); to study UTLS transport and mixing timescales (Hoor et al., 2010); to study UTLS ozone variability (Thouret et al., 2006; Manney et al., 2011; Cohen et al., 2018; Olsen et al., 2019); to intercompare CO measurements in the tropopause region (Martínez-Alonso et al., 2014); to assess the reanalyses representation of ozone mini-holes (Millán and Manney, 2017); to characterize the composition of the

Asian summer monsoon anticyclone (Santee et al., 2017); and to study cyclone-induced surface ozone and HDO depletion in the Arctic (Zhao et al., 2017).

The purpose of the present paper is two-fold: First, it introduces the OCTAV-UTLS SPARC activity and, second, it provides information and examples of the dynamical diagnostics to be used in future OCTAV-UTLS activity studies.

## 2 OCTAV-UTLS

As already mentioned, the OCTAV-UTLS SPARC activity aims to reduce uncertainties in UTLS composition trend estimates by accounting for dynamically induced variability (Kunkel et al., 2018; Hoor et al., 2019; Leblanc et al., 2020). The distribution of tracers (not only ozone) in the UTLS shows large spatial and temporal variability, caused by competing transport, chemical, and mixing processes near the tropopause, variations in the tropopause itself, as well as the position and variations of the jets (i.e., Hegglin et al., 2009; Manney et al., 2011). This strongly affects quantitative estimates of the impact of radiatively active substances, including ozone and water vapor, on surface temperatures (e.g., Forster and Shine, 1997; Riese et al., 2012), and complicates the diagnosis of dynamical and transport processes such as stratosphere-troposphere exchange (e.g., Gettelman et al., 2011). The community thus faces the challenge of optimally exploiting the vast existing portfolio of observations to better understand the physical composition of the UTLS.

Achieving OCTAV-UTLS goals requires a detailed characterization of existing measurements (from aircraft, ground-based, balloon, and satellite platforms) in the UTLS, including understanding how their quality and sampling characteristics (spatial and temporal coverage, resolution) affect the representativeness of these observations. Paramount to this effort is to develop and apply common dynamical diagnostics (described in this paper) to compare UTLS observations using a variety of geophysically-based coordinate systems (e.g., equivalent latitude, relative to the tropopause, or relative to the jets) derived from meteorological information from reanalysis datasets. This approach provides a framework for comparing and eventually combining measurements with diverse sampling patterns and thus leverages the meteorological context to derive maximum information on UTLS composition and its relationships to dynamical variability.

Ultimately, taking into account the trends in the tropopause or jet locations will lead to more accurate and comprehensive trace species trend estimates in the UTLS. Thus, after the comparison of measurements with different sampling characteristics, OCTAV-UTLS will facilitate studying long-term trends in chemical species from the various observational platforms.

## 3 Datasets

Within OCTAV-UTLS, ozonesonde, lidar, aircraft, and satellite datasets will be used. Ozonesondes are in-situ sensors normally flown attached to a balloon and interfaced with meteorological radiosondes (Komhyr, 1986; Komhyr et al., 1995). Recent ozonesondes are equipped with a GPS receiver. In short, the ozonesonde measures the current generated by the reaction of atmospheric ozone with a solution of potassium iodide, which is proportional to the ozone concentration.

The ozonesonde data used here were taken from the National Oceanic and Atmospheric Administration (NOAA) Global Monitoring Laboratory (GML) open-access archive. In the troposphere, typical one sigma uncertainties are around ±5% at midlatitudes and up to ±20% in the tropics; lower stratospheric uncertainties are around ±4–6% (e.g., Smit et al., 2007; Sterling et al., 2018; Tarasick et al., 2021). Their vertical resolution depends on the weight of the package, balloon size, weather conditions, and the sampling rate. Analog ozonesondes were able to sample approximately every 60 seconds and digital models can sample up to every second (Sterling et al., 2018), resulting in a vertical resolution varying from approximately 300 m to 5 m. Ozonesondes used in OCTAV-UTLS have been gridded to 250 m (for analog sondes) or 100 m (for digital sondes) to reduce computing power when calculating the dynamical diagnostics. In principle, ozonesonde records provide long-term UTLS ozone measurements under all weather conditions, that is, not biased toward clear-sky conditions. Table 1 lists the ozonesondes particulars.

Lidar (an acronym for Light Detection And Ranging) is a laser remote sensing technique commonly used to measure atmospheric composition from the ground, aircraft, and occasionally space. One or more laser beams are emitted into the atmosphere and a fraction of the emitted light is scattered back to the lidar instrument receiver by molecules and particles, where it is sampled as a function of time, i.e., distance traveled from the instrument. Tropospheric and stratospheric ozone can be measured using the differential absorption lidar (DIAL) technique, for which two laser beams of different wavelengths are emitted and absorbed differently by ozone along their atmospheric path. This difference in absorption is used to deduce ozone number density (Mégie et al., 1977). The typical vertical resolution of ozone DIAL measurements ranges from a few meters in the lower troposphere, to a few kilometers in the upper stratosphere. Precision typically ranges from less than 1% to over 10% depending on the altitude and vertical resolution considered (Leblanc et al., 2016). Table 1 also lists the lidars currently available.

Because of their technical complexity and cost, the number of ozone DIAL instruments is limited to less than a dozen around the world, thus implying limited geographical coverage. However, lidars have the capability to measure continuously for many hours, or even days without interruption (e.g. Godin-Beekmann et al., 2002; Leblanc et al., 2018), making the technique very well suited for a wide range of atmospheric studies ranging from long-term monitoring to physical processes in support of meteorology and air quality forecast and modeling (e.g., Ravetta et al., 2007; Steinbrecht et al., 2009; Cooper et al., 2010; Langford et al., 2018; Zerefos et al., 2018). The long-term stability of ground-based lidar instruments also makes them very suitable for validation of airborne or satellite-based measurements (e.g., Jiang et al., 2007; Gijsel et al., 2009; Hubert et al., 2016; Wang et al., 2020; Wing et al., 2020; Mettig et al., 2022). Due to their inherent measurement sampling characteristics, lidars' effective vertical and temporal resolutions can be optimized to specific applications, making the lidar technique very versatile in comparison to other observation platforms (e.g., Leblanc et al., 2012).

Aircraft in-situ measurements are typically made using UV photometry and/or chemiluminescence detectors (CLDs). UV photometry is based on the strong UV ozone absorption; simply, the light intensity measured by a photometer and the concentration of ozone in the absorption chamber are described by the Beer–Lambert law. CLDs use either a gaseous, solid or liquid reagent. The sensor measures the intensity of light emitted from a reaction product in an electronically excited state which is proportional to the ozone concentration. In particular, we use measurements from the following field campaigns:

- In-service Aircraft for a Global Observing System (IAGOS) Civil Aircraft for the Regular Investigation of the atmosphere Based on an Instrument Container (CARIBIC) (Brenninkmeijer et al., 1999, 2007)

- Spurenstofftransport in der Tropopausenregion (SPURT, which means trace gas transport in the tropopause region) (Engel et al., 2006)

- Stratosphere-Troposphere Analyses of Regional Transport (START08) (Pan et al., 2010)

- Transport and Composition in the Upper Troposphere and Lower Stratosphere and Earth System Model Validation (TACTS/ESMVal) (Müller et al., 2016)

- Polar Stratosphere in a Changing Climate (POLSTRACC) campaign (Oelhaf et al., 2019). Note that, POLSTRACC was operated with two other projects, the GW-LCYCLE (Investigation of the Life cycle of gravity waves) and SALSA (Seasonality of Air mass transport and origin in the Lowermost Stratosphere), known together as the PGS mission

- Wave-driven ISentropic Exchange (WISE) (Kunkel et al., 2019)

Typical random errors are smaller than 1% for newer systems (current CARIBIC measurements, Zahn et al. (2012)) but around 5% for older aircraft measurements (e.g., SPURT Hegglin et al. (2006)).

Aircraft measurements from commercial flights (IAGOS), although they are essentially line measurements limited in time and space, provide a very detailed view of UTLS ozone due to their high temporal resolution, relatively low measurement error, and long timeseries. Their 10 to 12 km cruising altitude coincides with the UTLS in mid and high latitudes. Research aircraft missions (such as SPURT, START08, WISE, etc) are limited in regional and temporal coverage, but cover a larger portion of the UTLS in the vertical and thus are better suited to process studies. Table 2 lists aircraft campaign particulars.

Satellite ozone measurements can be cataloged according to their measurement geometry, namely, nadir emission, nadir scattering, limb emission, limb scattering, and solar, lunar, or stellar occultation (more information in Chapter 2 of Hegglin and Tegtmeier, 2017). In particular, here we use solar occultation and limb emission measurements. Limb-viewing instruments generally yield a higher vertical resolution and larger sensitivity than nadir-viewing instruments in the stratosphere.

The solar occultation technique uses measurements of sunlight through the Earth's atmosphere and ratios them with ones where no atmospheric attenuation was present (i.e., a measurement above the atmosphere). Hence, the solar occultation technique is a self calibrated measurement. These measurements provide a very high signal-to-noise ratio (due to the brightness of the sun) allowing the detection of species with very low atmospheric concentrations. The main limitation of this technique is that the measurements can only be taken during sunrise or sunset, constraining the number of measurements to two per orbit. The solar occultation measurements used here are from the Atmospheric Chemistry Experiment Fourier Transform Spectrometer (ACE-FTS) and Stratospheric Aerosol and Gas Experiment III on the International Space Station (SAGEIII/ISS).

ACE-FTS was launched in 2003 on board the Canadian SCISAT spacecraft and measurements started in February 2004. It retrieves temperature, pressure, and concentration for several dozen atmospheric trace gases from infrared measurements (Bernath et al., 2005). ACE-FTS focuses on high-latitude science, and thus almost 50% of its occultations occur at latitudes

at or poleward of 60°. The nominal vertical resolution of this dataset (as defined by the field-of-view) is about 3 km but Hegglin et al. (2008) showed that ACE-FTS is capable of resolving the UTLS with high accuracy, suggesting a nominal vertical resolution of around 1 km, which is achieved thanks to oversampling. That is, measuring at vertical spacing finer than the instrument field-of-view of 3 km. Studies validating the ozone data include Dupuy et al. (2009), Sheese et al. (2017), and Sheese et al. (2021), among others. We have used ACE-FTS version 4.1.

SAGEIII/ISS was launched and delivered to the ISS in 2017 and routine measurements started in June 2017. It measures UV, visible, and near-IR sunlight through the Earth's limb to retrieve ozone, water vapor, nitrogen dioxide, and aerosol concentrations. The vertical resolution of this dataset is about 1 km. SAGEIII/ISS ozone measurements were validated by McCormick et al. (2020), Wang et al. (2020), and Hegglin et al. (2021). Here we have used SAGEIII/ISS version 5.2.

The limb emission viewing technique uses observations of thermal emission, that is, the radiation emitted by the atmosphere along a line of sight, to infer atmospheric constituents. These techniques provide day and night retrievals with dense sampling throughout the orbit. Limb emission measurements from the Aura Microwave Limb Sounder (MLS) are used here. MLS was launched onboard NASA's Aura satellite and measurements started in August 2004. It measures limb millimeter and submillimeter atmospheric thermal emission, from which temperature, trace gas concentrations, and cloud ice are retrieved (Waters et al., 2006). Aura MLS ozone has around 3 km resolution in the UTLS and stratosphere (Livesey et al., 2020), and the data have been extensively validated (e.g., Jiang et al., 2007; Froidevaux et al., 2008; Livesey et al., 2008; Hubert et al., 2016). Here we use MLS version 5.

Satellite observations, despite the coarser vertical and horizontal resolution in comparison with the other datasets here discussed, provide a global picture of the UTLS and can help with linking measurements with limited geometrical coverage. Table 3 lists these ozone satellite measurement characteristics including typical precision in the UTLS.

Figure 1 shows the sampling locations of the ozonesondes, lidars, aircraft, and satellite instruments used in OCTAV-UTLS. These datasets differ greatly in geographical coverage, horizontal and vertical resolution, measurement frequency, and time period covered. To fully exploit these measurements, careful attention needs to be paid to understand how sampling differences between the instruments can affect their representation of the atmospheric state (e.g, Hegglin et al., 2008; Toohey et al., 2013; Lin et al., 2015; Millán et al., 2016; Miyazaki and Bowman, 2017; Millán et al., 2018; Chang et al., 2020).

The datasets used here are not meant to be an exhaustive list of all the ozone records available; many other records exist. For example, the rest of the Network for the Detection of Atmospheric Composition Change (NDACC) ozonesonde records (e.g., Mazière et al., 2018), the ozonesondes included in the Southern Hemisphere ADditional OZonesondes (SHADOZ)(Witte et al., 2017; Thompson et al., 2017), the IAGOS-CORE (Petzold et al., 2015), or the limb scattering satellite sounders, the Optical Spectrograph and Infrared Imager System (OSIRIS) (Llewellyn et al., 2004) or the Ozone Mapping and Profiler Suite (OMPS) (Seftor et al., 2014). However, the records included in this study are meant to be representative of the currently available ozone datasets in terms of resolution as well as geographical sampling and with a focus on the extratropical UTLS, where tropopause and jet variability have their strongest impact on ozone gradients and variability.

## 4 Dynamical Diagnostics Methodology

In this section, we describe the dynamical diagnostics currently used in OCTAV-UTLS as well as the methodology for their generation. Figure 2 shows a timeline of the dynamical diagnostics available to date. Note that this timeline reflects the processing of the dynamical diagnostics and not the timeline of the measurement record, which in many cases extends to the current date.

Briefly, these dynamical diagnostics provide at each measurement time and location a comprehensive set of meteorological variables useful for characterizing the effects of dynamical variability on composition. The meteorological data used herein to compute the dynamical diagnostics are from the Modern-Era Retrospective Analysis for Research and Applications version 2 (MERRA-2) reanalysis (Gelaro et al., 2017). MERRA-2 is produced using version 5.12.4 of the Goddard Earth Observing System assimilation system using a 3D-FGAT (first guess at the appropriate time) with an incremental analysis update (Bloom et al., 1996) assimilation scheme to constrain the analyses (Lawless, 2010). The MERRA-2 dataset covers from 1980 to the present, providing fields every 3 h on a $0.625°$ by $0.5°$ longitude/latitude grid with 72 hybrid $\sigma$-pressure levels between the surface and 0.01 hPa. Its typical UTLS spacing is about 1.2 km. The approximate vertical resolutions for its entire vertical range can be found in Figure 3 by Fujiwara et al. (2017). Between January 1980 and September 2004, MERRA-2 assimilates partial column $O_3$ retrievals from the Solar Backscatter Ultraviolet Radiometer instruments and starting in October 2004, it assimilates $O_3$ profiles from Aura MLS and total column $O_3$ from the Ozone Monitoring Instrument (Wargan et al., 2017). MERRA-2 products have been extensively compared with those from other reanalyses and found to be well-suited for UTLS studies (e.g., Manney et al., 2017, 2021a, b; Homeyer et al., 2021; Fujiwara et al., 2022; Tegtmeier et al., 2022).

When computing dynamical diagnostics as discussed in this section it is important to use the same reanalysis fields for all the datasets to be used in a given study. This is to avoid introducing artifacts in the climatology and trend analyses from differences between the meteorological fields. For example, because of changes in assimilated datasets and different processing streams, discontinuities in the temperature and wind time series occur (Long et al., 2017). The magnitude and timing of these discontinuities vary between reanalyses and could confound the interpretation of studies using dynamical diagnostics derived from different reanalysis. Xian and Homeyer (2019) found when comparing WMO tropopauses derived from reanalysis fields with tropopauses derived from radiosonde data, that in general, the MERRA-2 primary tropopause altitudes have a positive bias while other reanalyses have negative biases. Millán et al. (2021) found, intercomparing dynamical tropopauses from multiple reanalysis fields, that root mean square daily altitude differences could be up to 1 km over most of the globe and greater than 2 km over Greenland, the Andes, Antarctica and around $30°S$ and $30°N$. Manney et al. (2017) showed comprehensive comparisons of five modern reanalyses for UT jets, the "subvortex" jet (see below), and multiple tropopauses, indicating generally good agreement in overall UT and subvortex jets characteristics, larger discrepancies in multiple tropopause characterization, and strong dependence of agreement in all diagnostics on the vertical grid resolution of the reanalyses. Manney and Hegglin (2018) evaluated long-term changes in UT jet latitude, altitude, and strength in the same reanalyses. While the signs of trends were usually the same, their magnitudes, uncertainty, and significance differed considerably, and some regions and seasons showed differences even in the signs of trends from different reanalyses in the southern hemisphere. Manney et al. (2021a)

diagnosed relationships of UT jet variability to El Niño Southern Oscillation (ENSO) using three modern reanalyses includ-ing MERRA-2, finding overall good agreement in the relationships between UT jets and ENSO among them. Manney et al. (2021b) used MERRA-2 and two other reanalyses to characterize the Asian summer monsoon anticyclone (ASMA), and found an altitude-dependent positive bias in MERRA-2 ASMA area with respect to the other reanalyses. A thorough reanalysis in-tercomparison can be found in the SPARC Reanalysis Intercomparison Project (S-RIP) Final Report (Fujiwara et al., 2022), of which Chapters 7 and 8 (Homeyer et al., 2021; Tegtmeier et al., 2022) focuses on the extratropical and tropical UTLS, respectively.

Thus, given these discontinuities, it is crucial to use the same reanalysis for all datasets throughout a given study. While using only one reanalysis can't completely eliminate the impact of these discontinuities, it can ensure that their effects are consistent across all datasets studied. That is, the timing of these discontinuities is well documented and it can be identified. If more than one reanalysis were used, the timing of discontinuities will be different, complicating the interpretation of such studies. That said, there is also value in repeating a study with meteorological fields from different reanalyses to study the sensitivity of the results to the uncertainties in the dynamical variables as recommended by S-RIP (Fujiwara et al., 2022).

The dynamical diagnostics are computed using the JEt and Tropopause Products for analysis and characterization (JETPAC) software. Originally, these algorithms were developed to process satellite measurements, but they have since been adapted to accommodate a diverse range of atmospheric measurements, including ozonesondes, lidars, and aircraft campaigns. Notably, this is the first time that all these records will be characterized consistently.

The JETPAC algorithms are described in detail by Manney et al. (2011, 2014, 2017, 2021a), and Manney and Hegglin (2018). For completeness, a review of the previously published use of JETPAC products for characterizing composition measurements is given here along with an update describing the current capabilities (for example, processing high resolution datasets) and their applications. This review also describes the JETPAC algorithm as applied to characterizing measurement environments, as opposed to referring the reader to Manney et al. (2011) that describes a previous iteration of the algorithms for two satellite datasets, and Manney et al. (2014, 2017, 2021a), and Manney and Hegglin (2018) that use JETPAC products only on the re-analysis grids for dynamical studies. In other words, the previous publications do not give a full view of the current capabilities of JETPAC for characterizing composition measurements.

The dynamical diagnostics generated by JETPAC can be broken down into 3 categories: derived meteorological products, tropopause characterization, and UT Jet and Subvortex jet identification. JETPAC is modular and other reanalyses besides MERRA-2 could be used in the future to verify that the results for measurement-location dynamical diagnostics are not re-analysis dependent. When computing dynamical diagnostics for multiple datasets it is also important to use the same code to compute such diagnostics. For example, other algorithms exist to derive equivalent latitude (e.g., Hoor et al., 2004; Hegglin et al., 2006; Añel et al., 2013), jets information (e.g., Strong and Davis, 2008; Spensberger and Spengler, 2020, and references therein) or tropopause information (e.g., Cohen et al., 2021; Homeyer et al., 2021, and references therein). All of these algo-rithms involve several assumptions, which are different in different algorithms, and also numerous interpolations which may be done differently. Thus, significant differences could result simply from using a different code for different datasets.

## 4.1 Derived meteorological products

The derived meteorological products are reanalysis fields, such as potential temperature ($\theta$), interpolated to the measurements' times and locations, as well as some derived meteorological products such as static stability (the gravitational resistance of the atmosphere to vertical displacements) and equivalent latitude (EqL). EqL is a quasi-Lagrangian coordinate defined as the geographical latitude encompassing the same area as the given potential vorticity (PV) contour (e.g., Butchart and Remsberg, 1986). EqL makes use of the conservation laws for potential vorticity for adiabatic frictionless flow. Under these conditions, PV is conserved and can be considered a passive tracer. In the stratosphere, this approximately holds for time scales of days to weeks. Here, tropospheric diabatic effects and turbulence do not play a significant role and the time scales are short enough that radiation and large scale downwelling do not dominate changes in PV. EqL is widely used in stratospheric studies (Brunner et al., 2006; Davis et al., 2017; Millán and Manney, 2017; Thomason et al., 2018; Manney et al., 2020, and references therein), as well as in UTLS studies to account for the local dynamical tropopause location (e.g., Strahan, 1999; Hoor et al., 2004; Engel et al., 2006; Hegglin et al., 2006; Bönisch et al., 2009; Krause et al., 2018). Table 4 lists the meteorological fields saved at each measurement location. The meteorological fields are interpolated linearly in time and bilinearly in latitude and longitude. Vertical interpolations are linear in $\log(\theta)$ for PV or $\log(\text{pressure})$ for other products. EqL is calculated on isentropic surfaces and interpolated linearly in $\log(\theta)$. The isentropic surfaces used are a grid commensurate with the MERRA-2 vertical spacing, from which EqL is then interpolated to the final desired levels (typically the potential temperatures of the measurement locations for the products described herein).

An important product from the reanalyses is the assimilated ozone at the measurement times and locations. This product not only facilitates studies of the reanalyses' representation of ozone, but also facilitates sampling biases studies by allowing comparison of the complete reanalysis fields mapped into a given coordinate system versus the assimilated ozone at the measurement locations in such coordinate system. For example, Chapter 7 of the S-RIP report (Homeyer et al., 2021) illustrates that sampling biases can be a confounding factor even with dense sampling such as from MLS. Conducting sampling bias studies can aid in evaluating the comparability of datasets with very dissimilar sampling patterns, including those presented herein (see Figure 1). Moreover, sampling differences can result in significant biases in the inferred magnitude of ozone trends, as highlighted by Millán et al. (2016).

As an example, Figure 3 shows the ozonesonde measurements (ozone, temperature, and wind speed) and MERRA-2 derived meteorological products (ozone, temperature, wind speed, and static stability) for the launches on 21 December 2009 and 12 January 2010 from Boulder, Colorado. On 21 December, MERRA-2 fields overestimate the ozone and wind speed while on 12 January, the derived meteorological products display excellent agreement with the ozonesonde measurements.

## 4.2 Tropopause characterization

An accurate depiction of the tropopause is important because it acts as a dynamical barrier separating strongly stratified stratospheric air from well-mixed tropospheric air, thus resulting in strong gradients in trace gases across the tropopause (e.g., Pan et al., 2004; Hoor et al., 2004; Hegglin et al., 2009). At each measurement time and location, JETPAC identifies the ther-

mal tropopause using the WMO definition (e.g., Homeyer et al., 2010), that is, where the temperature lapse rate falls below $2 \, \mathrm{K \, km^{-1}}$ for at least 2 km. We also use JETPAC to identify the dynamical tropopauses using four PV values (2, 3.5, 4.5, and 6 potential vorticity units). PV has been widely used to identify the tropopause location since stratospheric air possesses higher values of PV than tropospheric air and changes in static stability result in strong vertical PV gradients at the tropopause (e.g.,

Morgan and Nielsen-Gammon, 1998). Since PV does not provide a well-defined tropopause near the tropics (e.g., Holton et al., 1995), these dynamical tropopauses are commonly defined by an isentropic surface (typically 380 K) wherever the PV contour definition would place it at a higher potential temperature (e.g., Schoeberl, 2004; Manney et al., 2011). The four PV values used for these dynamical diagnostics span the range of most commonly used values in the literature (e.g., Hoskins et al., 1985; Highwood et al., 2000; Schoeberl, 2004; Randel et al., 2007; Kunz et al., 2011a; Pan et al., 2012), and are consistent with

PV-gradients in the extratropics that arguably best reflect the dynamical barrier and tracer gradients (Kunz et al., 2011a, b).For the calculations herein, JETPAC searches for tropopauses above 1.5 km and for pressure levels smaller than 600 hPa, to avoid identifying boundary layer inversions as tropopauses.

In addition to the primary tropopause, multiple tropopauses are also identified by JETPAC above the primary tropopause every time their particular definition is fulfilled. Note that to identify multiple thermal tropopauses, we only require the temper-

15 ature lapse rate to drop below $2 \, \mathrm{K \, km^{-1}}$ as opposed to $3 \, \mathrm{K \, km^{-1}}$ as defined by the WMO (WMO, 1957) . Randel et al. (2007) showed that the frequency of double thermal tropopause occurrence is dependent on the chosen lapse rate threshold, with $2 \, \mathrm{K \, km^{-1}}$ resulting in better agreement when comparing double tropopauses derived from (relatively coarse resolution) reanalyses with those derived from high resolution datasets using the strict WMO definition. Multiple dynamical tropopauses are also reported, and physically represent regions of tropopause folding, which are often associated with stratosphere-troposphere

exchange.

In addition to the derived meteorological products, Figure 3 also displays the location of the 4.5 PVU and WMO tropopauses. On 21 December 2009, there was only one tropopause associated with the profile (at 204 and 216 hPa for the 4.5 PVU and WMO tropopause, respectively), while on 12 January 2010, two WMO tropopauses were identified by JETPAC (at 201 and 98 hPa), coincident with strong gradients of static stability. On 12 January 2010, the locations of the primary WMO and

25 4.5 PVU tropopauses are almost the same. This figure also shows tropopause inversions (e.g., Birner et al., 2002, 2006). The tropopause inversion layer (TIL) is a shallow layer just above the tropopause, corresponding to a local maximum in static stability. These inversions often occur in the same regions and seasons as multiple tropopauses (e.g. Grise et al., 2010; Peevey et al., 2014; Schwartz et al., 2015). The representation of the TIL showcases the ability of modern reanalyses to capture the sharp gradients associated with such inversions (e.g., Gettelman and Wang, 2015; Kedzierski et al., 2016; Wargan and Coy,

2016). The ability to properly represent these inversion layers is important to define coordinates capable of capturing the trace gas gradients in the UTLS.

For many of the composition datasets, the latitude, longitude, and measurement time change with pressure/altitude, for example, during an ozonesonde measurement, the balloon drifts from the launch location, or for a limb measurement the lat/lon position of the tangent height may change significantly as the satellite orbits the Earth while the instrument scans the

35 atmosphere. As such, in principle, we could provide tropopause information at every single measurement point along the

balloon ascent / satellite measurements; however, this may generate a large amount of mostly redundant information, and can result in very large files, so we instead provide just one tropopause characterization.

Figure 4 shows examples of how we determine a unique tropopause for a balloon ascent as well as for satellite scans whose locations vary significantly with altitude.

5    – First, we compute the tropopause information (based on the interpolated reanalysis fields) for the measurements below 30 km as their lat / lon change with height along the balloon ascent or along the limb scan (gray dots in Fig. 4).

   – Then, we compute the mean of the primary tropopauses computed (indicated by purple arrows in Fig. 4).

   – This mean tropopause value typically doesn't coincide exactly with a measurement altitude (best seen in the ACE-FTS panel in Fig. 4 because of the coarser retrieval grid of its measurements).

10    – We thus find the measurement location where the tropopause value is closest to the mean value just computed (indicated by red arrows in Fig. 4). The tropopause characterization for that level (and hence geographic location) is selected as the most representative tropopause characterization for the entire balloon ascent or the entire limb scan.

   – We repeat this procedure for each tropopause definition.

To explore the tropopause characterization further, Figure 5 shows maps of the mean WMO tropopause altitude for several 15 datasets, as well as the double WMO tropopause occurrence frequency. The representation of the altitude of the WMO primary tropopause and the WMO double tropopause frequency depends on the time and sampling patterns of the measurements. The most noticeable examples are the high values of double tropopause occurrence (above 80%) in the CARIBIC-2 dataset in the Northern Atlantic Ocean close to the US coast and the 20-30% double tropopause occurrence around Greenland only present for the MLS measurements despite CARIBIC-2 and SAGEIII/ISS measuring at least part of that region. The list of tropopause 20 characteristics saved at each measurement location can also be found in Table 4.

## 4.3   UT Jet and subvortex jet identification

At each measurement longitude, upper tropospheric jet cores are identified in latitude/vertical coordinate slices where the reanalysis wind speed maximum exceeds $40\,\mathrm{ms}^{-1}$. The jet boundaries are the grid points surrounding that core (both vertically and horizontally) where the wind speed drops below $30\,\mathrm{ms}^{-1}$. When there is more than one wind speed maximum greater than 25   $40\,\mathrm{ms}^{-1}$ within a given $30\,\mathrm{ms}^{-1}$ contour, they are identified as separate cores if the minimum wind speed between them is at least $30\,\mathrm{ms}^{-1}$ less than the wind speed value at the strongest core, or if the latitude distance between them is greater than $15°$. These criteria were chosen to approximate the selections that would be made by visual inspection. As an example, following Manney et al. (2011), Figure 6 illustrates the jet identification. This figure shows two cross-sections of wind speed, one in Southern Hemisphere spring and one in the Northern Hemisphere winter. In both cross sections there are two $40\,\mathrm{ms}^{-1}$ maxima 30   within a given $30\,\mathrm{ms}^{-1}$ contour. In the Southern Hemisphere, they are cataloged as one jet core (identified as the subtropical jet (STJ)) while in the Northern Hemisphere they are cataloged as two jet cores, (N1 and polar jet (PJ)).

In each hemisphere, JETPAC identifies the subtropical jet as the jet core closer to the equator for which the WMO tropopause altitude at the equatorial edge is greater than 13 km and for which the tropopause altitude is at least 2 km lower on its opposite (poleward) side. The polar jet is defined as the strongest jet core poleward of the subtropical jet or poleward of $40°$ latitude if no subtropical jet is identified. These criteria ensure that, in general, the identified subtropical jets are the ones close to the tropopause break (i.e., the abrupt drop in the height of the WMO tropopause while transitioning from the tropics to midlatitudes), consistent with primarily radiative driving, while the polar jets are consistent with primarily eddy driving. Any other jet cores (up to five at the longitude/time of a given measurement) are also cataloged; the catalog is ordered by decreasing jet core windspeed, with subtropical and polar jets indicated by a flag in the dynamical diagnostic files.

In each hemisphere, JETPAC identifies the subvortex jet core as the most poleward westerly reanalysis wind speed maximum, as long as it exceeds $30\,\mathrm{ms}^{-1}$, at each model level above near 300 hPa (the white dots in Figure 6). The subvortex edges are the location of the $30\,\mathrm{ms}^{-1}$ windspeeds on both sides of the subvortex core (i.e., poleward and equatorward, the blue dots in Figure 6). Between 300 and 80 hPa, the bottom of the subvortex jet is distinguished from the top of an upper tropospheric jet as the lowest altitude at which the wind speed of the jet is still decreasing with decreasing altitude. Once the upper tropospheric jets and subvortex jet are identified, we catalog the distances (vertically in pressure, height and potential temperature and horizontally in latitude) from them to the measurement locations. Table 4 lists all the jet and subvortex parameters saved from JETPAC.

As we did for the tropopause identification, we provide just one jet and subvortex identification per balloon ascent / limb scan. This identification is similar to the unique tropopause identification explained before, and it is computed as follows: First, we compute the UT jet and subvortex jet information for all measurements through a balloon ascent or limb scan. Then, we compute the mean jet core wind speed of the strongest UT jet throughout those calculated through the locations along a balloon ascent or limb scan. The mean jet core wind speed typically doesn't coincide exactly with the jet core wind speed at any of the measured locations, so we identified the measured location with jet core wind speed closest to that mean jet core wind speed. The information at this measured location is taken to be the best representation of the UT jet and subvortex jet information for the entire balloon ascent or limb scan.

# 5  Coordinate systems

The dynamical diagnostics discussed here are the first comprehensive and consistent set of diagnostics across multi-platform datasets. In addition to allowing mapping of the datasets into additional conventional coordinate systems (for example, the MLS pressure-based dataset into altitude or the SAGEIII/ISS altitude-based dataset into pressure), these diagnostics allow mapping of such datasets into several dynamically based latitude-like horizontal coordinates and altitude-like vertical coordinates (Manney et al., 2011). Potentially useful horizontal coordinates include latitude, equivalent latitude, and latitude distance from the subtropical or polar jet. Vertical coordinates include altitude, pressure, potential temperature, altitude or potential temperature from a jet core, and altitude or potential temperature from a tropopause. Table 5 shows specific horizontal and vertical coordinate combinations enabled by the dynamical diagnostics.

Mapping into a given coordinate system can be performed by simply averaging the composition measurements within coordinate bins. Variability within a given bin can be characterized by standard deviations, median, minimum/maximum values, percentiles, etc. The number of points in each bin can be easily stored to ensure an adequate number of data points within the bins and to facilitate combining mapped fields into longer averaging periods. Figure 7 displays various ozone datasets mapped using latitude versus altitude, a commonly utilized coordinate system. These datasets were selected to showcase the variety of records included on OCTAV-UTLS: MLS and SAGEIII/ISS showcase satellite measurements with dense and sparse sampling, CARIBIC-2 illustrates aircraft measurements, while the Boulder and Table Mountain measurements represent the ozonesonde and lidar records, respectively. As expected, ozone is low and well mixed in the troposphere, increasing in the lower stratosphere due to photochemical production.

The sampling of each record is evident in the zonal means presented here. MLS, with its dense sampling and near-global coverage (82°S-82°N), provides a comprehensive perspective of global ozone fields, capturing the climatological view of the subtropical jet and subvortex jet (black contours, see Figure 6 for reference). However, MLS suffers from a lack of coverage below ∼10 km and coarser vertical resolution compared to the other datasets used here. On the other hand, SAGEIII/ISS offers sampling in the upper troposphere with measurements down to 5 km, with better vertical resolution, but is limited to measurements between 60°S and 60°N. The distorted wind contours in comparison to MLS illustrate the greater impact of sampling biases than for MLS.

CARIBIC-2 measurements are restricted to flight levels, mostly situated at altitudes below 12 km, near the extratropical tropopause. These measurements, due to their high frequency observation rate, capture much finer details that are not resolved by satellite products. As such, it provides an exceptional perspective for studying tropopause-related processes in this region, but its coverage is limited elsewhere. Finally, ozone measurements obtained by ozonesondes and lidars provide superior vertical resolution, but are confined to their specific measurement locations. To enhance their visibility, we have replicated the ozonesonde and lidar measurements in the adjacent latitude bins in Figure 7-top.

The different sampling of each of these datasets may contribute to biases / artifacts and confound the interpretation of the results. To better understand this issue, OCTAV-UTLS aims to characterize the sampling biases of these datasets by comparing ozone reanalysis fields that are interpolated to the measurement times and locations (included in the dynamical diagnostics discussed herein) with the raw reanalysis fields (similar to the approach used in previous studies e.g., Toohey et al. (2013); Millán et al. (2016)).

Figure 7 also shows the datasets mapped into latitude versus altitude with respect to the WMO tropopause. Tropopause coordinates allow for the separation of tropospheric and stratospheric air and remove variability caused by differences in tropopause height, allowing better characterization not only of troposphere to stratosphere gradients, but also of errors associated with each regime. It has been shown that using tropopause coordinates greatly reduces the scatter around the tropopause, revealing a sharp gradient between tropospheric and stratospheric air masses (e.g., Pan et al., 2004; Hoor et al., 2004; Hegglin et al., 2006, 2009). This coordinate system has been used to validate ozone measurements from several satellite instruments against coincident measurements (e.g., Monahan et al., 2007; Pittman et al., 2009). Furthermore, Hegglin et al. (2008) argue that this coordinate system can be used in a climatological manner, enabling the inclusion of all available measurements as op-

posed to just spatio-temporally coincident ones, thus improving validation statistics especially for instruments with less dense sampling patterns.

The impact on coverage of the tropopause mapping can be seen in particular in MLS and in CARIBIC-2. Although MLS shows very few measurements below the tropopause in the extratropics in the latitude/altitude view, mapping with respct to the tropopause allows separation of individual measurements that were taken above and below the tropopause. This shows that MLS does on occasion sample below the extratropical tropopause. Similarly, in the extratropics, CARIBIC-2 covers only a few kilometers above the tropopause in the latitude/altitude mapping, but the use of tropopause coordinates shows that it covers air masses that are high above the local tropopause (e.g., above deep folds) and thus samples deep stratospheric air.

Figure 7 also shows the datasets mapped into equivalent latitude versus potential temperature coordinates. The benefits of using EqL/$\theta$ coordinates (or PV/$\theta$) for data comparisons are well established (e.g., Hegglin et al., 2006; Lait et al., 2004; Velazco et al., 2011). Trace gases with transport timescales significantly shorter than their chemical timescales are well mixed along EqL (or PV) contours on isentropic surfaces (e.g., Leovy et al., 1985), which facilitates comparisons of non-coincident measurements taken in the same air mass, in addition to providing criteria to filter out coincident measurements that were taken in different air masses (a common occurrence in regions of strong PV gradients such as the stratospheric vortex edge or the tropopause) (e.g., Michelsen et al., 2002; Lumpe et al., 2002; Chiou et al., 2004; Lumpe et al., 2006; Velazco et al., 2011; Griffin et al., 2017; Ryan et al., 2017; Bognar et al., 2019).

All instruments show expanded "condition space" coverage when mapped in EqL/$\theta$ coordinates (e.g., Schoeberl et al., 1995; Manney et al., 2009, and references therein); this is particularly noticeable for the ozonesonde and lidar datasets, which, while physically measuring at a single location, show a broad distribution of ozone values typical of a wide range of dynamical conditions found throughout the respective hemisphere they are located in. Figure 8-left depicts a normalized count of the ozonesonde and lidar coverage in equivalent latitude. The vertical dashed lines display the geographical latitude for the ozonesonde launches or the lidar locations (noting that geographical and equivalent latitude have completely different meanings and implications). Any deviations observed between these lines and the maximum normalized count (i.e., 1) indicate that the respective datasets are situated in a region of significant dynamical variability, which leads to diabatic PV modification and thus a shift in equivalent latitude, such as the South Pole and Summit, Greenland ozonesondes, or the Table Mountain lidar measurements. As shown, the expanded space can cover from 11 to 30° (determined as the half width of the normalized counts). Note that, in Figure 7, all other records show coverage improvements, with SAGEIII/ISS covering most of the globe, CARIBIC-2 extending its coverage throughout the northern hemisphere, and MLS covering the entire globe.

However, studies have shown that PV or EqL coordinates must be used with caution in the upper troposphere because adiabatic PV-conservation is violated particularly by phase transitions of water as well as regional turbulence (in particular in the vicinity of the jet cores) and a lack of a single simply connected approximately circumpolar transport barrier (e.g., Manney et al., 2011; Pan et al., 2012; Jin et al., 2013; Kaluza et al., 2021). In this region, other coordinate systems may map ozone differently and thus provide additional insights for interpreting the data. Figure 7 shows the datasets mapped into latitude with respect to the STJ versus altitude with respect to the WMO tropopause. This coordinate combination almost completely segregates the ozone measurements taken in different air masses by separating tropospheric and stratospheric air

in the vertical and air on either side of the transport barrier represented by the subtropical jet in the horizontal. This separation is most evident in the CARIBIC-2 panel. As discussed by Manney et al. (2011), the horizontal jet coordinate reorders the data geometrically and thus highlights ozone, tropopause height, and strong local horizontal PV gradients across the jets in the latitude region strongly influenced by those jets; EqL (or PV) provides a complementary view that may not highlight the strongest PV gradients because of regional variations in the strength and position (in EqL) of the jet, but helps identify various physical processes influencing the trace gas distributions, such as mixing related to diabatic motions, and does, as mentioned above, also help in distinguishing tropospheric and stratospheric air.

Note that when referring to latitude with respect to the STJ, all instruments show an expanded measurement space, which is again most noticeable for the ozonesonde and lidar datasets. MLS, with its denser sampling, has full sampling of the region near the subtropical jets, as indicated by the concentric wind contours (black line) around the zero line. Figure 8-right also shows the extent of this coverage for the ozonesondes and lidars.

In addition to the composition measurements, we can map derived meteorological products such as winds, temperature, ozone, PV, potential temperature, static stability, etc into the same dynamically-based coordinates. Figure 9 is an example of this type of mapping, showing MERRA-2 temperature fields interpolated to the measurement times and locations for several datasets mapped in the same coordinates shown in Figure 7. Sampling biases are more obvious for temperature. Differences up to 7 K can be seen in comparing the ozonesonde and lidar datasets (when using EqL and latitude wrt STJ coordinates) with MLS data. MLS has relatively dense sampling (see Figure 1), which should minimize its sampling biases (Toohey et al., 2013; Millán et al., 2016) at least in the stratosphere; nevertheless, Homeyer et al. (2021) showed that even that relatively dense MLS sampling resulted in significant biases in assimilated ozone fields with respect to the reanalysis fields on their native high-resolution latitude longitude grids. As was the case for ozone, to fully understand these differences, the sampling biases need to be characterized. Figures A1, A2, A3, and A4, show examples of other datasets.

## 6 Summary and Future directions

Understanding long-term changes in UTLS ozone is exceptionally challenging, in part because of the impact of the spatial and temporal variability of the jet streams and tropopause locations that are instrumental in controlling trace gas distributions. As part of the SPARC OCTAV-UTLS activity, we have generated dynamical diagnostics for multi-platform (ozonesondes, lidars, aircraft campaigns, and satellite) ozone records that allow such datasets to be mapped consistently into dynamical coordinates.

These dynamical diagnostics are computed with the well-established JETPAC software (Manney et al., 2011, 2014, 2017; Manney and Hegglin, 2018). They can be broken down into three categories:

- Derived meteorological products: Meteorological fields interpolated to the measurement times and locations.

- Tropopause Characterization: Tropopause information (either thermal or dynamical, as well as primary, secondary and tertiary, etc) at the measurement times and locations.

– UT Jet and SubVortex jet Identification: UT Jet and subvortex jet cores are identified and cataloged in latitude/height slices at each measurement longitude.

Mapping multi-platform datasets into different coordinate systems will help us understand processes controlling UTLS ozone regionally and globally, as well as at seasonal and longer time scales. Suitable dynamical coordinates, such as tropopause coordinates and jet based coordinates will help us separate air masses with distinct characteristics, such as the tropospheric air where ozone is low and well mixed and stratospheric air where it increases with height. Further, using dynamical coordinates allows separation of variability arising from different mechanisms, such as dynamical versus chemical processes.

Future OCTAV-UTLS studies on ozone will:

– Investigate which dynamical coordinates better homogenize the mapped measurements, that is, which ones best segregate air in regions separated by geophysical transport barriers (for example, the tropopause) and thus provide the most comparable results within a given bin for non-coincident measurements. A paper on this topic is already on preparation. An illustrative example is shown on Figure 10-a for the WISE campaign using the relative standard deviation (RSTD, i.e., the standard deviation divided by the mean) as metric to assess the variability in different coordinate systems. As expected, the dynamical coordinates display a smaller RSTD.

– Explore how sampling biases can confound the intercomparability of these records. These biases will be explored by comparing the ozone reanalysis fields interpolated to the measurement times and locations of the dynamical diagnostics discussed here to the raw reanalysis fields. Additionally, this study will explore which coordinates can, if any, reduce sampling biases. As an example, Figure 10-b shows the MLS sampling biases for January 2005.

– Analyze the impact of dynamical coordinates on quantification of long-term trends. The aim is to identify regions where dynamical coordinates can reduce the uncertainty associated with such trends. As an example, Figure 10-c shows MLS 20°S-20°N time series at 100 hPa as well as at the WMO tropopause. As shown, the time series at 100 hPa displays much larger amplitude variations that are related to the annual cycle of the 100 hPa pressure surface with respect to the tropopause, which results in sampling different fractions of stratospheric and tropospheric air in different seasons. Interannual differences in these variations are also likely related to changes in the relative altitudes of the 100 hPa isobaric surface and the tropopause.

Uncertainties arising from differences between reanalysis fields from different centers will also be examined by comparing the results when using different reanalyses as recommended by the SPARC Reanalysis Intercomparison Project. Other datasets, such as more ozonesondes, IAGOS-CORE, the Gimballed Limb Observer for Radiance Imaging of the Atmosphere (GLORIA), the Michelson Interferometer for Passive Atmospheric Sounding (MIPAS), the Optical Spectrograph and InfraRed Imager System (OSIRIS), and the Ozone Mapping Profile Suite (OMPS) may also be processed with JETPAC for use in studies that combine the most comprehensive set of measurements and thus maximize spatio-temporal coverage. Some of these additional datasets may be particularly useful when analyzing trace gases that are not measured by as many different platforms as ozone.

Ultimately, these dynamical diagnostics will allow satellite, aircraft, balloon, and ground-based observations of different long-lived and also shorter-lived trace gases to be studied in a physically consistent way in order to understand the impact of the physical processes (whether chemical or dynamical) driving trace-gas variability on the observed UTLS trends as viewed by the different sensors on different platforms. The dynamical diagnostics and mapping techniques discussed herein can be used with all other trace gases measured by any platform; water vapor and carbon monoxide are among the species that will be a future focus of OCTAV-UTLS studies. For example, within OCTAV-UTLS, Jeffery et al. (2022) constructed water vapor and ozone climatologies using equivalent latitude and potential temperature tropopause-relative coordinates in an effort to best represent the distribution of these gases in the UTLS.

*Data availability.* The ozone datasets used are available as follows:

– OzoneSondes: https://gml.noaa.gov/aftp/data/ozwv/Ozonesonde/

– Lidar: https://www-air.larc.nasa.gov/missions/ndacc/data.html

– SPURT: https://iacweb.ethz.ch/spurt

– WISE: https://halo-db.pa.op.dlr.de/

– START08: https://data.eol.ucar.edu/master_lists/generated/start08/

– TACTS/ESMVal: https://halo-db.pa.op.dlr.de/

– PGS: https://halo-db.pa.op.dlr.de/

– CARIBIC-1 and 2: https://www.caribic-atmospheric.com/Data.php

– ACE-FTS: http://www.ace.uwaterloo.ca

– ACE-FTS quality information: https://dataverse.scholarsportal.info/dataset.xhtml?persistentId=doi:10.5683/SP2/BC4ATC

– Aura MLS: https://disc.gsfc.nasa.gov/

– SAGEIII/ISS: https://asdc.larc.nasa.gov/

For the dynamical diagnostics please contact Gloria L Manney (manney@nwra.com) or Luis F Millán (lmillan@jpl.nasa.gov).

*Author contributions.* GLM wrote most of the core JETPAC software. LMV generalized the code to cope with several type of datasets. LMV wrote the paper. All the co-authors commented on and edited the manuscript. IP provided the ozonesonde datasets, TL provided the LIDAR datasets, PH and DK provided the SPURT, WISE, TACTS/ESMVal, and PGS datasets, AZ and HB provided the CARIBIC-1 and CARIBIC-2 datasets.

*Competing interests.* There are no competing interests.

*Acknowledgements.* LMV's and TL's research was carried out at the Jet Propulsion Laboratory, California Institute of Technology, under a contract with the National Aeronautics and Space Administration (80NM0018D0004). GLM was supported by subcontracts from JPL through the Microwave Limb Sounder (MLS) project and the SAGEIII/ISS project. PH and DK acknowldege support by the German Science foundation (DFG) by the TR 301 (Project 428312742). IP's research was supported by the NOAA Cooperative Agreement with CIRES, NA17OAR4320101. We thank the JPL MLS team (especially Brian Knosp and Ryan Fuller) for data management and processing support, and William Daffer for work on early development of JETPAC. MERRA-2 is an official product of the Global Modeling and Assimilation Office at NASA GSFC, funded by Modeling Analysis and Prediction. The Atmospheric Chemistry Experiment is a Canadian-led mission primarily supported by the CSA.

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

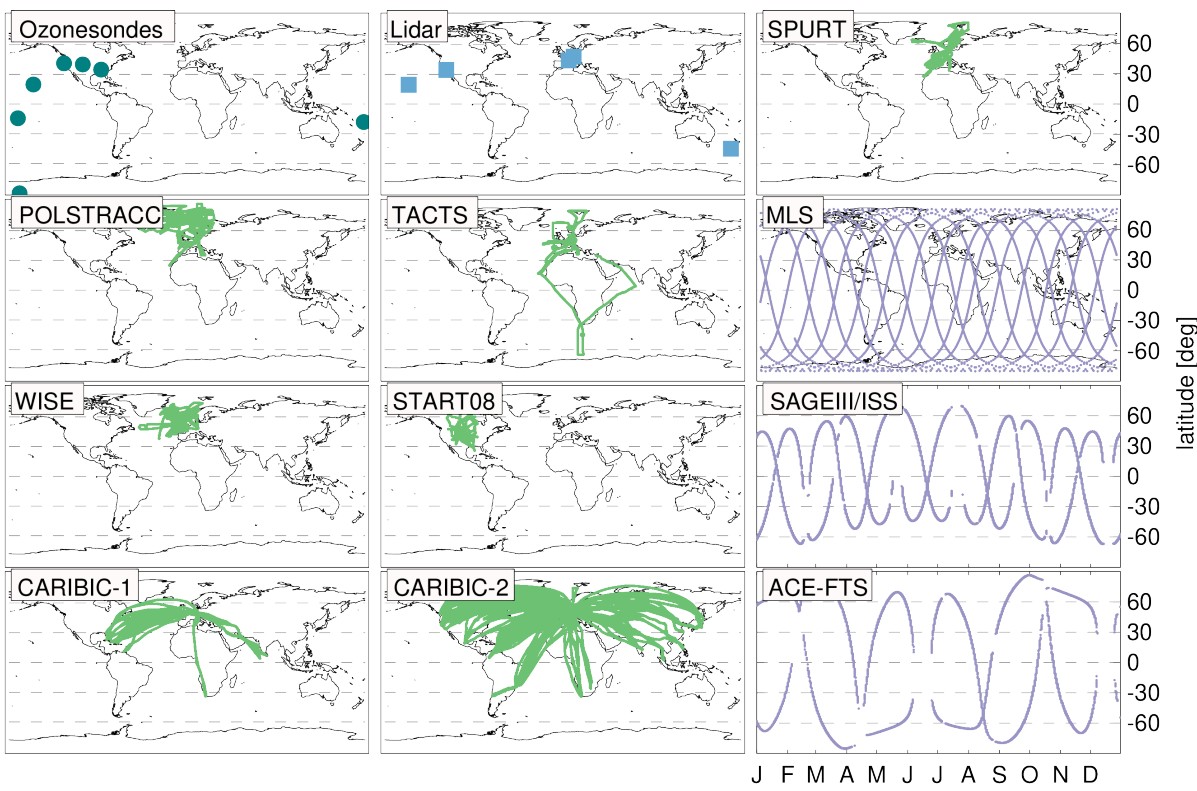

**Figure 1.** Locations of ozonesondes (teal circles) and lidars (blue rectangles), sampling of aircraft campaigns (light green), and representative sampling patterns for satellite instruments (MLS daily sampling pattern, SAGEIII/ISS and ACE-FTS yearly sampling patterns, all in purple).

Zhao, X., Weaver, D., Bognar, K., Manney, G., Millán, L., Yang, X., Eloranta, E., Schneider, M., and Strong, K.: Cyclone-Induced Surface Ozone and HDO Depletion in the Arctic, Atmospheric Chemistry and Physics Discussions, 2017, 1–35, https://doi.org/10.5194/acp-2017-427, 2017.

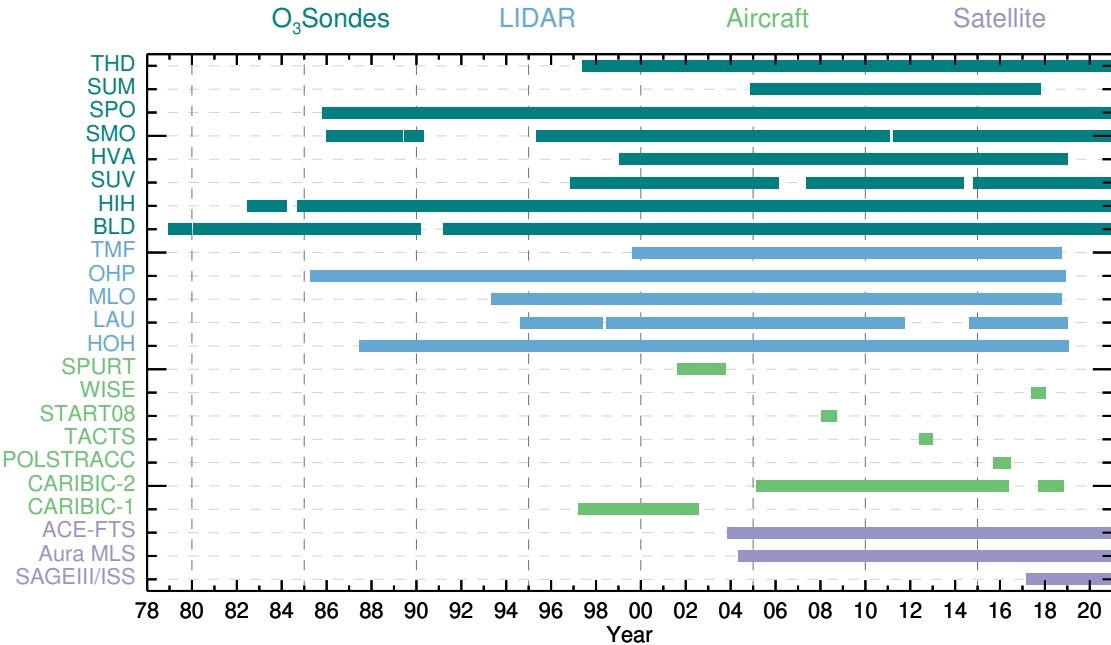

**Figure 2.** Dynamical diagnostics availability between 1979 - 2020 for the datasets considered within the OCTAV-UTLS SPARC activity.

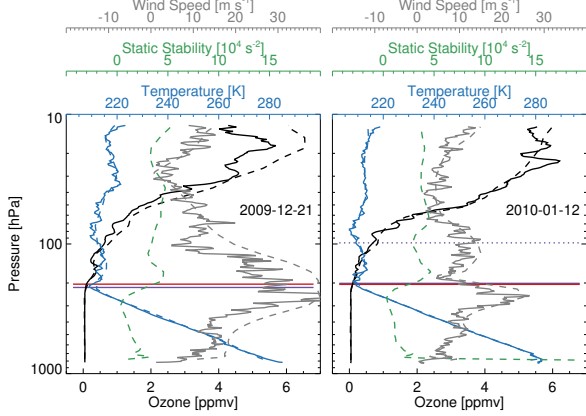

**Figure 3.** Ozonesonde measurements (solid) and MERRA-2 derived meteorological products (dashed) for the launches in 21 December 2009 and 12 January 2010 from Boulder, Colorado. Red lines show the 4.5 PVU dynamical tropopause, and purple lines the WMO (thermal) tropopause (dotted purple lines show the secondary thermal tropopause).

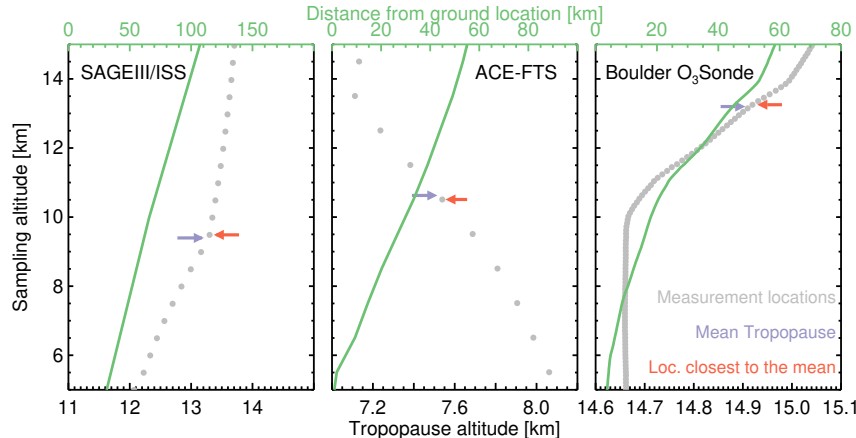

**Figure 4.** SAGEIII/ISS, ACE-FTS, and ozonesonde examples of the primary WMO tropopause variation through the balloon ascent / satellite measurements. The WMO tropopause characterization used is the one from the measurement location closest to the average tropopause through the balloon ascent / satellite measurements. An analogous procedure is used to identify unique dynamical tropopauses through the balloon ascent / satellite measurements. For satellites, distance from the ground location refers to the distance from the lowest available retrieval level.

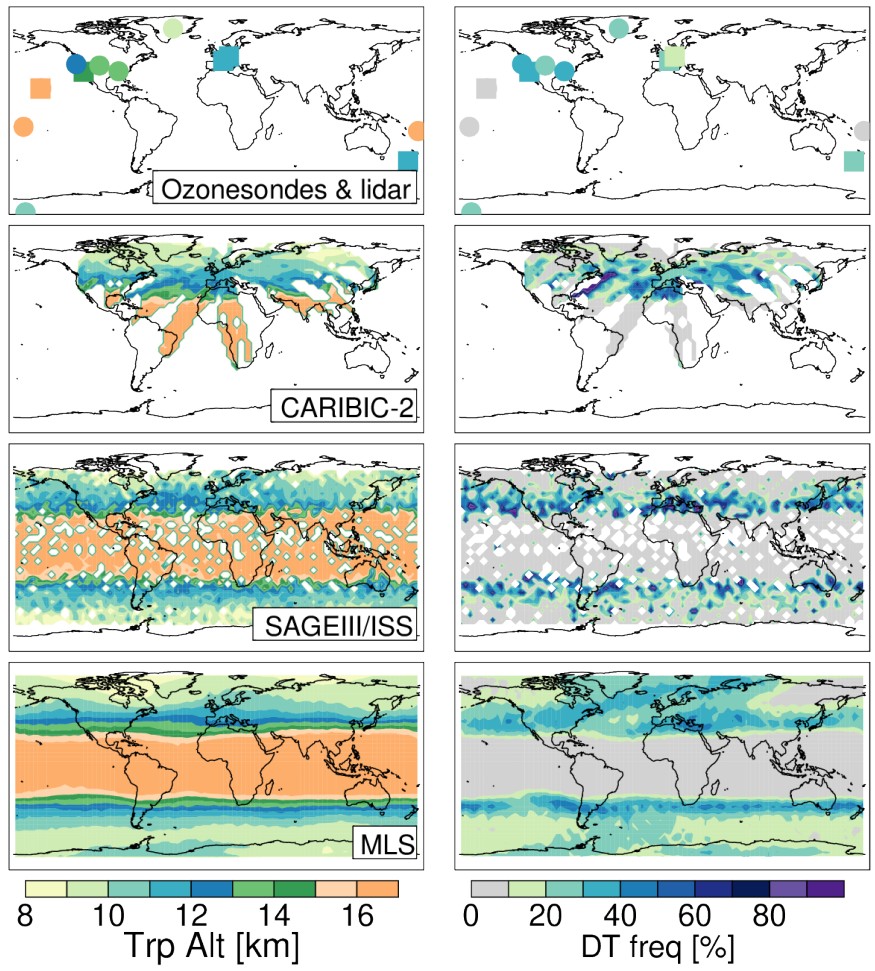

**Figure 5.** (left) Climatological MERRA-2 WMO primary tropopause altitude sampled at ozonesondes (circles), lidar (rectangles), CARIBIC-2, SAGEIII/ISS, and MLS measurements (right) Climatologies of MERRA-2 WMO double tropopause occurrence frequencies for the same datasets. MLS and SAGEIII/ISS display 2018 data, CARIBIC-2, Ozonesonde and Lidar show data post 2005.

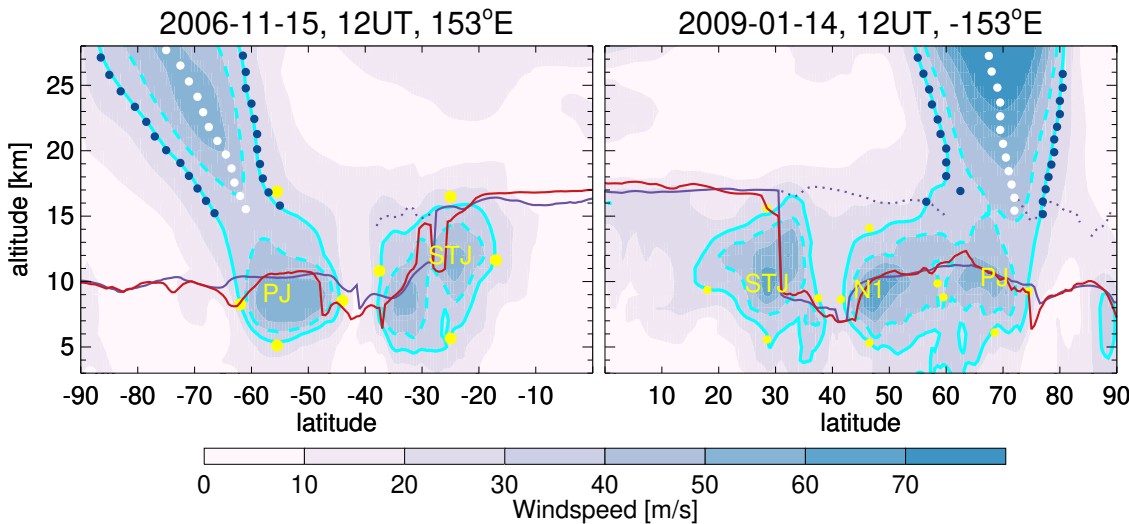

**Figure 6.** Cross-sections of MERRA-2 wind speed with jet and tropopause classification information overlaid. Yellow letters/numbers indicate the locations of jet cores; lowest numbers are for strongest jets in each hemisphere; yellow dots indicate the identified locations of the edges of the jet region (at grid-points, thus not exactly matching contours). Red line show the 4.5 PVU dynamical tropopause, and purple lines the WMO (thermal) tropopause (dotted purple lines show the secondary thermal tropopause). White dots show the subvortex jet maximum, and blue dots the edges of the subvortex jet. Figure based on Manney et al. (2011).

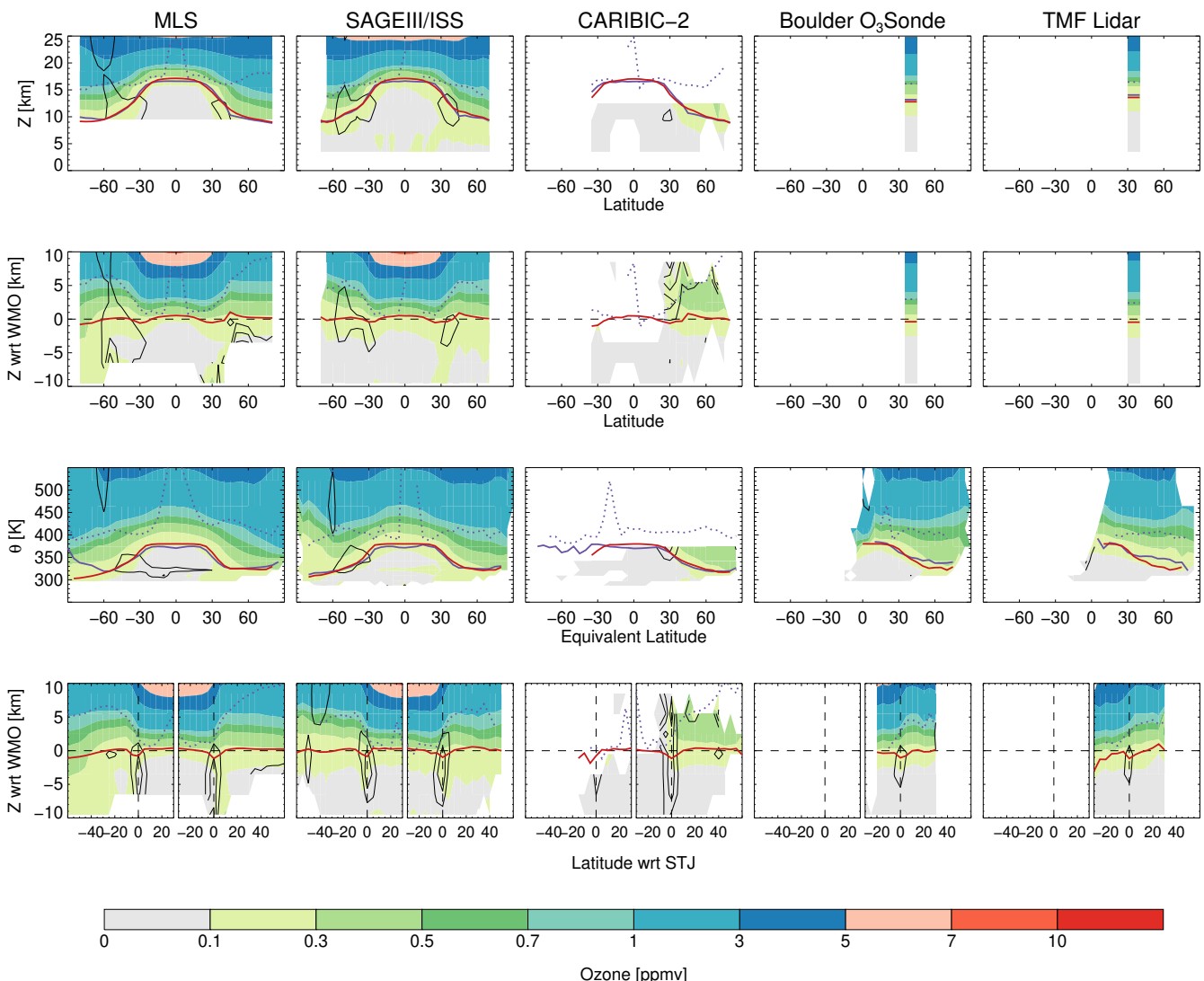

**Figure 7.** Climatological MLS, SAGEIII/ISS, CARIBIC-2, Boulder ozonesonde, and Table Mountain tropospheric ozone lidar data in different coordinate systems. MLS and SAGEIII/ISS display 2018 data, CARIBIC-2, Boulder Ozonesonde and Table Mountain Lidar show data post 2005. Red lines show the 4.5 PVU dynamical tropopause, and purple lines the WMO (thermal) tropopause (dotted purple lines show the secondary thermal tropopause). The black contours show wind speed values of 30, 40, and $50\,\mathrm{ms}^{-1}$. Note that differences in their representation in comparison with MLS suggest sampling biases. Datasets were binned on a $5°$ horizontal grid, and a 5 km or 5 K grid. Note that in the top two rows, the ozonesondes and lidar measurements (that only sample one latitude bin) have been replicated in the adjacent latitude bins to improve visibility.

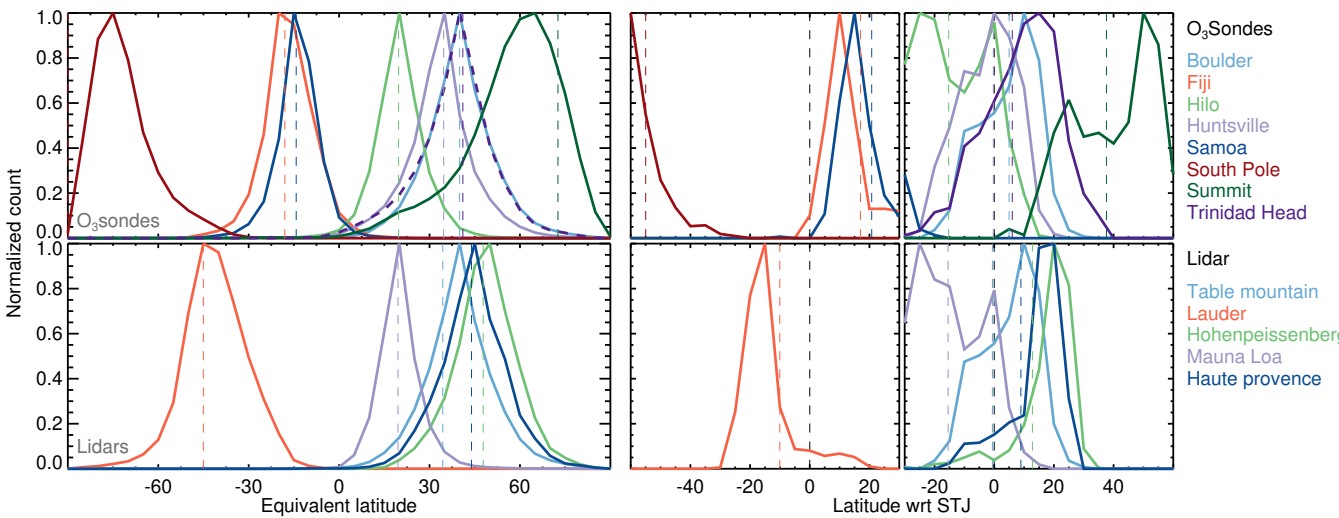

**Figure 8.** (left) Equivalent latitude range covered by the ozonesondes (top) and lidars (bottom). Vertical dashed lines display the actual latitude position for each dataset as listed in table 1. (right) Range covered by the same datasets but in latitude with respect to the subtropical jet. Vertical color dashed lines display the dataset latitude with respect to a STJ climatological position of -35°S and 35°N. A vertical black line indicate the position of the subtroical jet (i.e., 0) in this coordinate system. Note that this normalized counts include information from multiple heights, the coverage for specific heights could differ slightly.

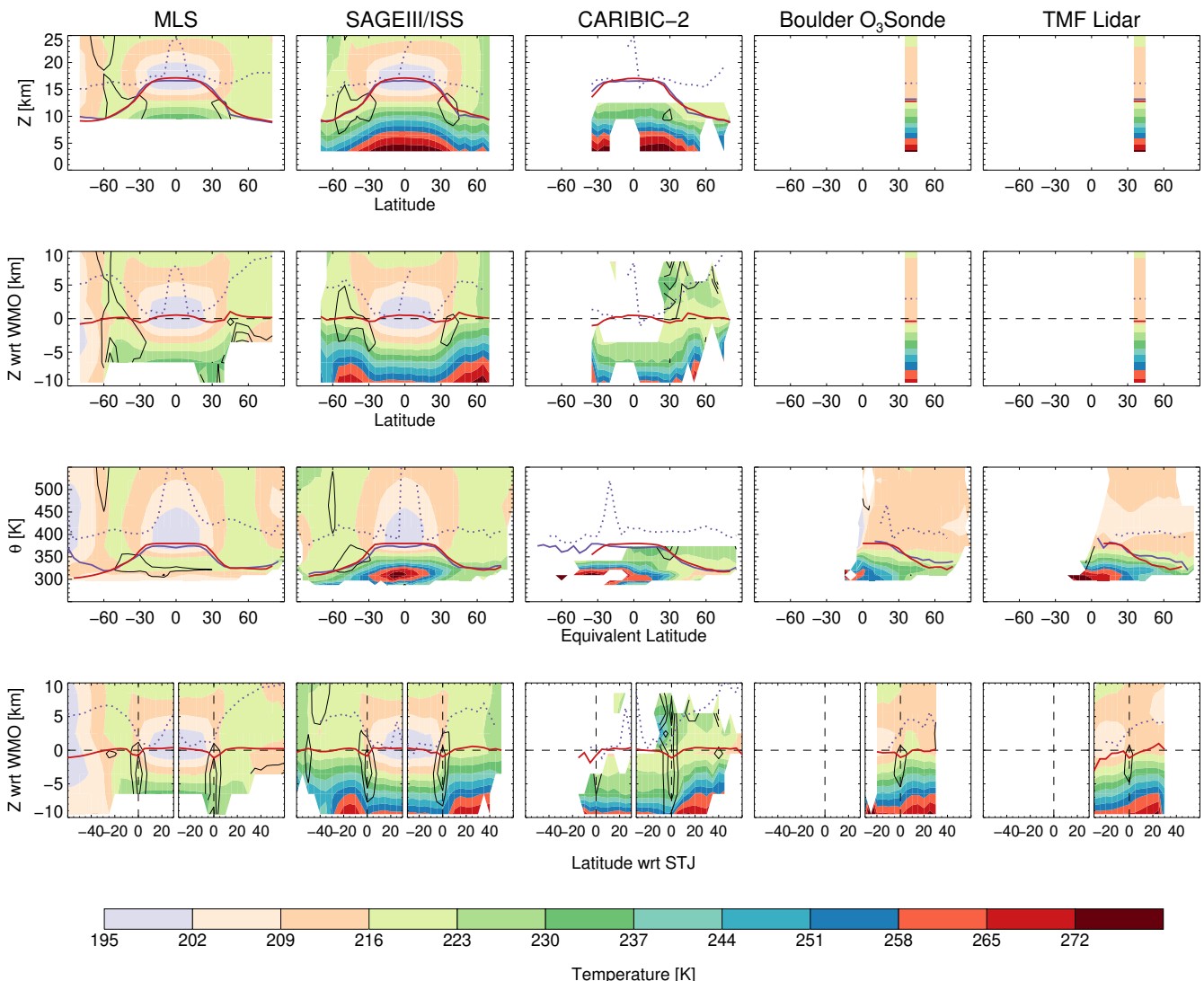

**Figure 9.** MERRA-2 temperature sampled at MLS, SAGEIII/ISS, CARIBIC-2, Boulder ozonesonde, and Table Mountain lidar measurements in different coordinate systems. MLS and SAGEIII/ISS panels display 2018 data, CARIBIC-2, Boulder Ozonesonde and Table Mountain Lidar panels show data post 2005. Red lines show the 4.5 PVU dynamical tropopause, and purple lines the WMO (thermal) tropopause (dotted purple lines show the secondary thermal tropopause). The black contours show wind speed values of 30, 40, and 50 ms$^{-1}$. Note that differences in their representation in comparison with MLS suggest sampling biases. Datasets were binned on a 5° horizontal grid, and a 5 km or 5 K grid.

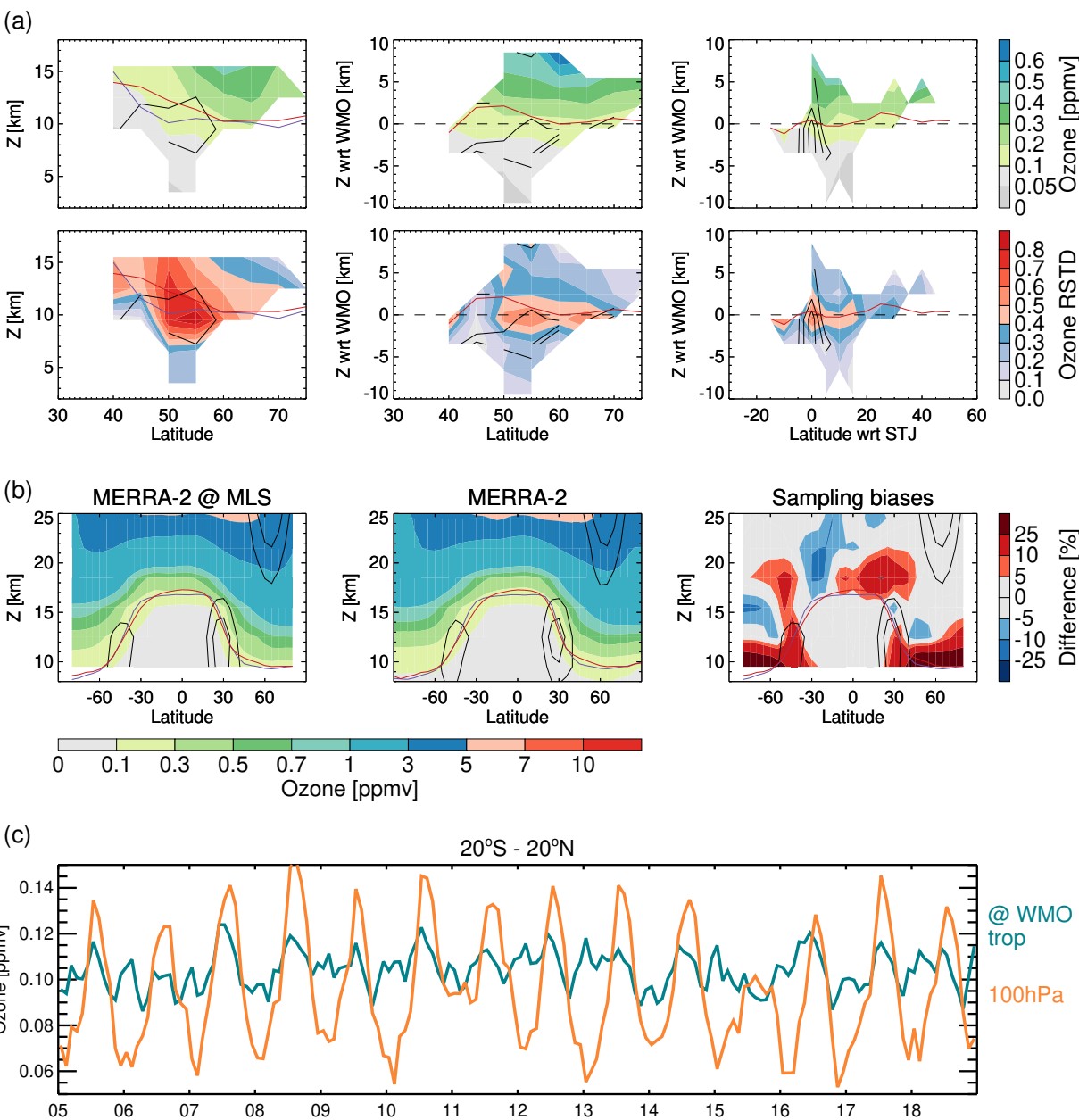

**Figure 10.** Illustrative examples of future OCTAV-UTLS studies. (a) WISE ozone data mapped into different coordinate systems (top) and their corresponding relative standard deviation (bottom). (b) January 2005 sampling biases as a function of latitude and altitude as computed using MERRA-2 fields interpolated at the MLS locations (i.e., one of the fields included in the dynamical diagnostics discussed here) versus the raw MERRA-2 fields. In (a) and (b), red lines show the 4.5 PVU dynamical tropopause, purple lines the WMO (thermal) tropopause, and black contours show wind speed values of 30, 40, and 50 ms$^{-1}$. (c) MLS time series at 20°S-20°N at 100 hPa and at 0 km wrt to the WMO tropopause.

**Table 1.** Details of ozonesondes (top) and lidar (bottom) datasets processed and their primary characteristics.

| Code | Name | Latitude | Longitude | Range | Timespan |
|---|---|---|---|---|---|
| SPO | South Pole, Antarctica | −89.98 | −24.8 | 0-30 km[a] | 1967-1971, 1986- |
| SUV | Suva, Fiji | −18.00 | 178.0 | 0-30 km | 1997- |
| SMO | Tutuila, American Samoa | −14.24 | −170.56 | 0-30 km | 1986-1990, 1995- |
| HIH | Hilo, Hawaii, USA | 19.72 | −155.05 | 0-30 km | 1982- |
| HVA | Huntsville, Alabama, USA | 34.720 | −86.64 | 0-30 km | 1999- |
| BLD | Boulder, Colorado, USA | 39.99 | −105.26 | 0-30 km | 1967-1971, 1979- |
| THD | Trinidad Head, California, USA | 41.05 | −124.15 | 0-30 km | 1997- |
| SUM | Summit, Greenland | 72.60 | −38.42 | 0-30 km | 2005-2017 |
| TMF | Wrightwood, California, USA | 34.4 | −117.7 | 10-50 km | 1989- |
| TMF[b] | Wrightwood, California, USA | 34.4 | −117.7 | 0-22 km | 1999- |
| OHP | Haute Provence, France | 43.94 | 5.71 | 10-50 km | 1985- |
| OHP[c] | Haute Provence, France | 43.94 | 5.71 | 10-50 km | 1991- |
| HOH | Hohenpeissenberg, Germany | 47.80 | 11.02 | 10-50 km | 1987- |
| MLO | Mauna Loa, Hawaii, USA | 19.54 | −155.58 | 10-50 km | 1993- |
| LAU | Lauder, New Zealand | −45.04 | 169.68 | 10-50 km | 1994- |

[a] For all ozonesondes, the highest altitude depends on the bursting point of the balloon

[b] There are two different lidars at TMF, a stratospheric system (measuring since 1989) and tropospheric one (measuring since 1999).

[c] There are two different lidars at OHP, a stratospheric system (measuring since 1985) and tropospheric one (measuring since 1991).

**Table 2.** Details of aircraft datasets processed and their primary characteristics.

| Campaign | Region | Timespan | Technique | References |
|---|---|---|---|---|
| CARIBIC-1 | N. Hemisphere | 1997-2002 | UV photometry | Brenninkmeijer et al. (1999) |
| CARIBIC-2 | N. Hemisphere | 2005-2020 | UV photometry | Brenninkmeijer et al. (2007) |
| SPURT | Europe | 2001-2003 | CLD and UV photometry | Engel et al. (2006) |
| START08 | Continental US | 2008 | CLD and UV photometry | Pan et al. (2010) |
| TACTS/ESMVAL | Europe and Africa | 2012 | CLD and UV photometry[a] | Müller et al. (2016) |
| PGS | Arctic | 2015-2016 | CLD and UV photometry[a] | Oelhaf et al. (2019) |
| WISE | Europe and N. Atlantic | 2017 | CLD and UV photometry[a] | Kunkel et al. (2019) |

[a] These campaigns all used the FAIRO instrument (Zahn et al., 2012).

**Table 3.** Satellite instrument datasets processed and their primary characteristics.

| Instruments | Timespan | # profiles per day | vert. res. | range | UTLS error | References |
|---|---|---|---|---|---|---|
| Aura MLS | 2004- | ∼3500 | ∼3 km | 316-0.01 hPa | ∼30% (UT) >4% (LS) | Waters et al. (2006) Livesey et al. (2020) |
| ACE-FTS | 2004- | ∼30 | 3-4 km | 5-90km | 3% | Bernath et al. (2005); Dupuy et al. (2009) |
| SAGEIII/ISS | 2017- | ∼30 | ∼0.5 km | 5-60 km | 3% | Wang et al. (2020) |

**Table 4.** Dynamical diagnostic products.

| Derived Meteorological Products[a] | |
|---|---|
| Pressure | Temperature |
| Zonal Wind | Meridional Wind |
| Potential Temperature | Geopotential Height |
| Altitude | Potential Vorticity |
| Scaled PV | Relative vorticity |
| Lapse Rate | Static stability |
| Equivalent Latitude[1] | Normalized Horizontal (Isentropic) PV Gradient[1] |
| Horizontal (isobaric) Temperature Gradient | Montgomery Stream Function[2] |
| Montgomery Stream Function | Assimilated Ozone |
| **Tropopause Characterization[b]** | |
| Number of tropopauses | Altitude |
| Pressure | Potential Vorticity |
| Potential Temperature | Static Stability |
| Lapse Rate | |
| **Jet Identification[c]** | |
| Jet location (lat / lon) | Jet types |
| Pressure | Temperature |
| Zonal Wind | Meridional Wind |
| Potential vorticity | Relative vorticity |
| Lapse Rate | Static stability |
| Potential Temperature | |

[a] Variables saved at each measurement location.

[1] Calculated on isentropic surfaces.

[2] Computed using the approximation $MSF = gz + c_p T$ where $g$ is earth's gravity, $z$ is the height of the isentropic surface, $c_p$ is the specific heat of air at constant pressure, and $T$ is the atmospheric temperature.

[b] Information saved at each tropopause (either thermal or dynamical as well as primary, secondary, tertiary, etc) location.

[c] Information saved at the jets maximum and edges (poleward, equatorward, inner, outer).

**Table 5.** Coordinate systems enabled by the dynamical diagnostics.

| | Altitude ($z$) | Pressure ($p$) | Potential temperature ($\theta$) | $z$ relative to tropopause | $\theta$ relative to tropopause | $z$ relative to STJ | $\theta$ relative to STJ | $z$ relative to PJ | $\theta$ relative to PJ |
|---|---|---|---|---|---|---|---|---|---|
| Latitude | • | • | • | • | • | • | • | • | • |
| Equivalent latitude | • | • | • | • | • | • | • | • | • |
| Latitude from STJ | • | • | • | • | • | • | • | | |
| Latitude from PJ | • | • | • | • | • | | | • | • |

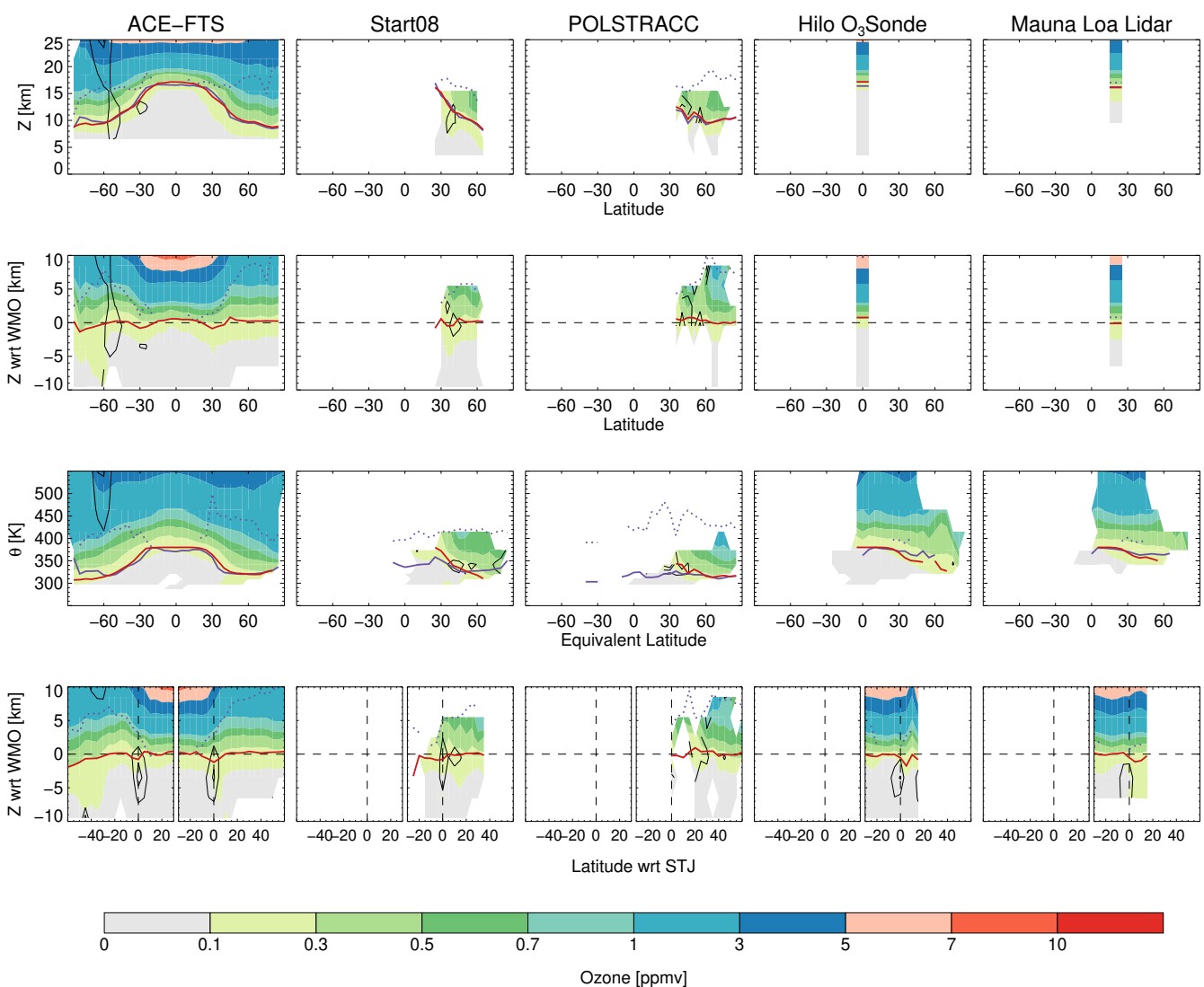

**Figure A1.** Climatological ACE-FTS, START08, POLSTRACC, Hilo ozonesonde, and Mauna Loa ozone data in different coordinate systems. ACE-FTS panel displays 2018 data, Hilo Ozonesonde and Mauna Loa Lidar show data post 2005, START08 and POLSTRACC display all their available data. Red lines show the 4.5 PVU dynamical tropopause, and purple lines the WMO (thermal) tropopause (dotted purple lines show the secondary thermal tropopause). The black contours show wind speed values of 30, 40, and $50\,\mathrm{ms}^{-1}$. Note that differences in their representation in comparison with MLS suggest sampling biases. Datasets were binned on a $5°$ horizontal grid, and a $5\,\mathrm{km}$ or $5\,\mathrm{K}$ grid.

## Appendix A: Other Datasets

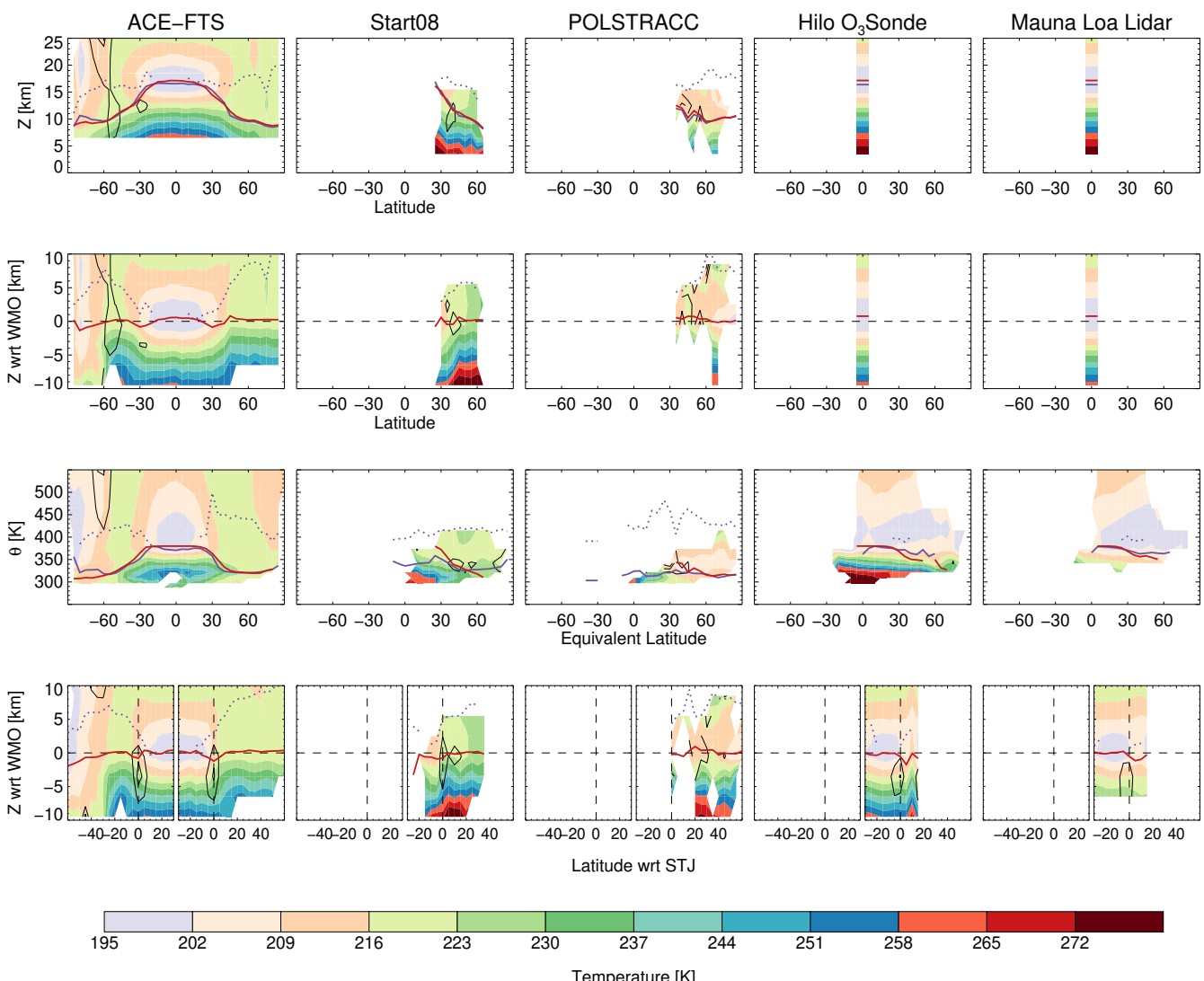

**Figure A2.** MERRA-2 temperature sampled at ACE-FTS, START08 POLSTRACC, Hilo ozonesonde, and Mauna Loa lidar measurements in different coordinate systems. ACE-FTS panel displays 2018 data, Hilo Ozonesonde and Lauder Lidar show data post 2005, START08 and POLSTRACC display all their available data. Red lines show the 4.5 PVU dynamical tropopause, and purple lines the WMO (thermal) tropopause (dotted purple lines show the secondary thermal tropopause). The black contours show wind speed values of 30, 40, and 50 ms$^{-1}$. Note that differences in their representation in comparison with MLS suggest sampling biases. Datasets were binned on a 5° horizontal grid, and a 5 km or 5 K grid.

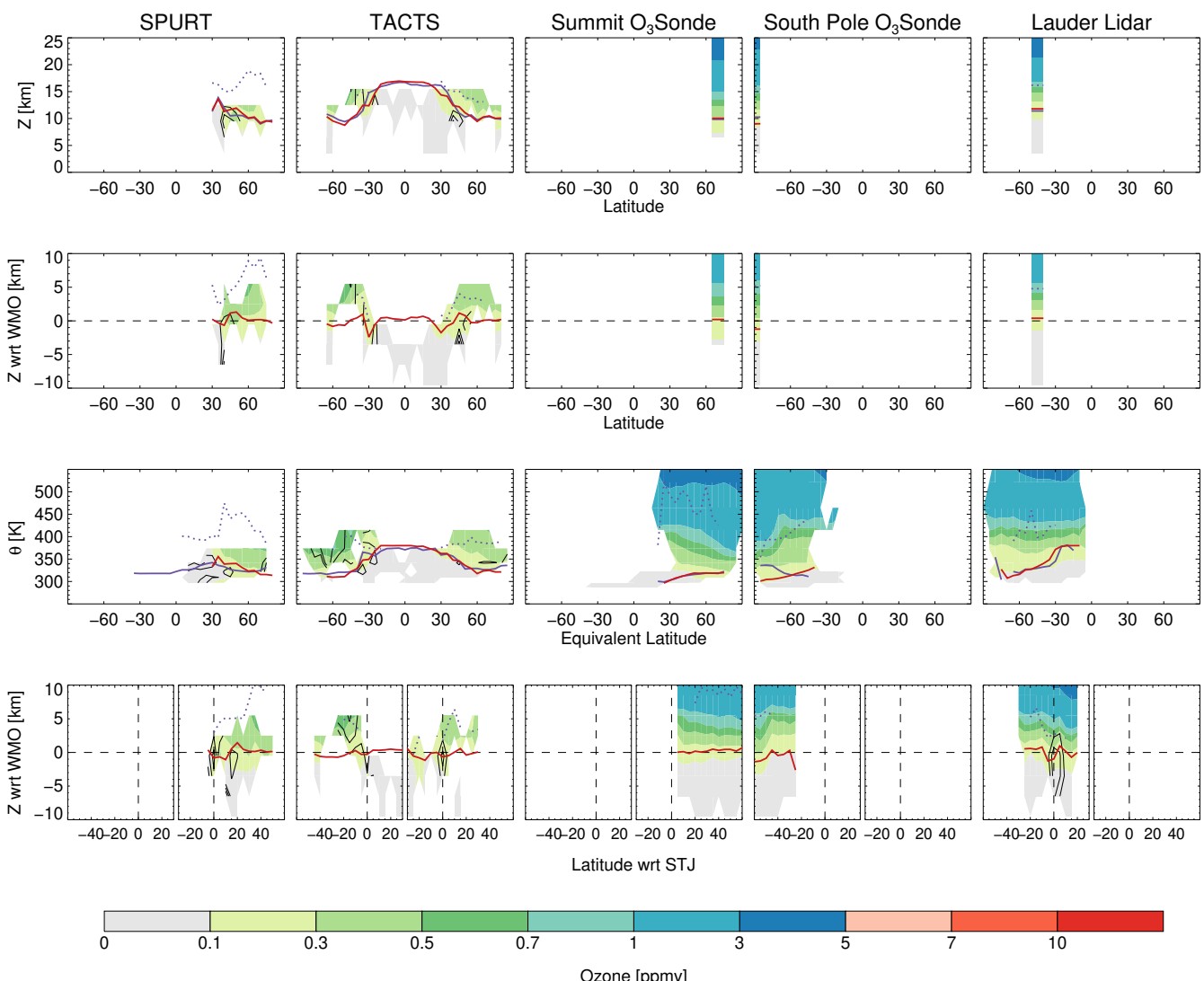

**Figure A3.** Climatological SPURT, TACTS, Summit and South pole ozonesondes, and Lauder Lidar ozone data in different coordinate systems. Summit, South Pole Ozonesonde and Lauder Lidar show data post 2005, SPURT and TACTS display all their available data. Red lines show the 4.5 PVU dynamical tropopause, and purple lines the WMO (thermal) tropopause (dotted purple lines show the secondary thermal tropopause). The black contours show wind speed values of 30, 40, and $50\,\mathrm{ms}^{-1}$. Note that differences in their representation in comparison with MLS suggest sampling biases. Datasets were binned on a $5°$ horizontal grid, and a $5\,\mathrm{km}$ or $5\,\mathrm{K}$ grid.

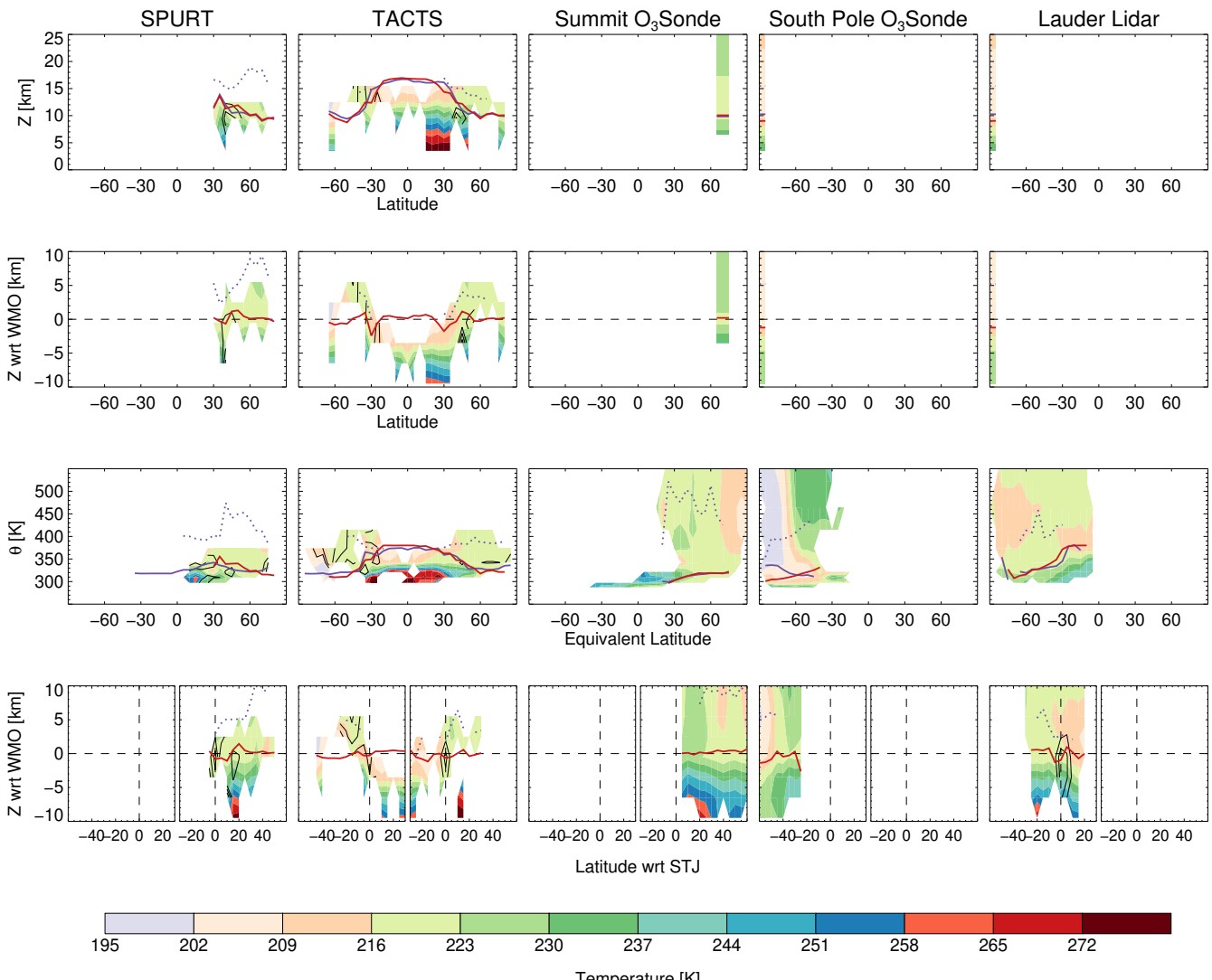

**Figure A4.** MERRA-2 temperature sampled at SPURT, TACTS Summit and South Pole ozonesondes, and Lauder lidar measurements in different coordinate systems. Summit and South Pole Ozonesonde and Lauder Lidar show data post 2005, SPURT and TACTS display all their available data. Red lines show the 4.5 PVU dynamical tropopause, and purple lines the WMO (thermal) tropopause (dotted purple lines show the secondary thermal tropopause). The black contours show wind speed values of 30, 40, and 50 ms$^{-1}$. Note that differences in their representation in comparison with MLS suggest sampling biases. Datasets were binned on a 5° horizontal grid, and a 5 km or 5 K grid.