# Peer review of "Multi-Parameter Dynamical Diagnostics for Upper Tropospheric and Lower Stratospheric Studies"

_EGUsphere, 2023_

## Author Comment (AC1)

We thank the reviewer for her/his comments. Below are our responses in blue.
Here is a summary of the major changes in the revised manuscript:

- We enlarged Figure 4, 5, 7, 8, A1 and A2 to improve readability.
- Figure A1 and A2 instead of showing Lauder lidar now show Mauna Loa so it can be compared against Hilo ozonesonde.
- We added two more figures in the appendix to highlight other datasets (SPURT, TACTS, Summit and South Pole ozonesondes and the Lauder lidar measurements).
- We change the title to: Multi-Parameter Dynamical Diagnostics for Upper Tropospheric and Lower Stratospheric Studies.
- We added a figure showing the range covered by the ozonesondes and lidars in EqL and latitude wrt STJ.
- We added discussions on the differences on sampling.
- We expanded the discussion of reanalysis discontinuities
- We emphasized the future OCTAV-UTLS studies with a figure showing some illustrative examples.

Reviewer 2 suggest that the manuscript is not suitable for this journal. The authors feel the material is appropriate for AMT, but if, after the revisions, the consensus among the reviewers and editor is that it is not, we will consider resubmitting (or transferring if that is possible) to ESSD. In any case, we thank the editor and reviewers in advance because we feel the paper is more complete after their suggestions.

**Review by Reviewer 1**
In this manuscript, the authors describe a set of multiple dynamical parameters for diagnostic studies of the upper troposphere – lower stratosphere (UTLS). This region is characterized by strong gradients in e.g. ozone, water vapor, or temperature in both the vertical and horizontal directions, and the use of the proposed dynamical parameters allows to better separate tropospheric and stratospheric air masses as well as air masses from either side of the subtropical or subvortex jets. This distinction help reduce uncertainties in monthly averages or in the comparison of measurements with different sampling patterns. Trends in tropopause altitude or subtropical jet position can also influence atmospheric parameter trends in this region. While the use of these dynamical parameters is not new, the manuscript provides an integrated view of their use and a detailed description of their calculation. The article is well written and documented. I thus recommend publication provided that the following comments are taken into account in a revised version.
**Main comments**
1. The benefit of using the diagnostic framework proposed in the manuscript for e.g. measurement comparison or trend evaluation is not clearly highlighted in the article. Figures 7 and 8 show some climatology comparison for ozone and temperature in the various coordinate systems but the advantage of using them is mainly based on citations (e.g. in page 13) while some demonstrations could be made in the article itself by showing for instance improved comparison time series or reduced error bars in monthly averages after using alternative coordinate systems.
   The reviewer is correct in pointing out that we could have shown some demonstrations of reduced error bars (i.e., due to correctly accounting for ozone gradients) or timeseries. However, those are the topics for the next OCTAV-UTLS studies as described in the summary and future directions section.

To emphasize this, that section was rewritten as shown below. We also included illustrative examples (Figure 10 in the current manuscript), but we note that the actual analysis of those topics is left for future papers.

Future OCTAV-UTLS studies on ozone will:
– Investigate which dynamical coordinates better homogenize the mapped measurements, that is, which ones best segregate air in regions separated by geophysical transport barriers (for example, the tropopause) and thus provide the most comparable results within a given bin for non-coincident measurements. A paper on this topic is already on preparation. An illustrative example is shown on Figure 10-a for the WISE campaign using the relative standard deviation (RSTD, i.e., the standard deviation divided by the mean) as metric to assess the variability in different coordinate systems. As expected, the dynamical coordinates display a smaller RSTD.
– Explore how sampling biases can confound the intercomparability of these records. These biases will be explored by comparing the ozone reanalysis fields interpolated to the measurement times and locations of the dynamical diagnostics discussed here to the raw reanalysis fields. Additionally, this study will explore which coordinates can, if any, reduce sampling biases. As an example, Figure 10-b shows the MLS sampling biases for January 2005.
– Analyze the impact of dynamical coordinates on quantification of long-term trends. The aim is to identify regions where dynamical coordinates can reduce the uncertainty associated with such trends. As an example, Figure 10-c shows MLS 20S-20N time series at 100 hPa as well as at the WMO tropopause. As shown, the time series at 100 hPa displays much larger amplitude variations that are related to the annual cycle of the 100 hPa pressure surface with respect to the tropopause, which results in sampling different fractions of stratospheric and tropospheric air in different seasons. Interannual differences in these variations are also likely related to changes in the relative altitudes of the 100 hPa isobaric surface and the tropopause.

The authors employ their diagnostic framework to some measurement time series e.g. a few ground-based sondes and lidar records, the MLS, SAGE III ISS and CARIBIC-2 time series and some campaigns measurements. It is not clear why these measurements were selected. In addition, some of the records are mentioned (e.g. SPURT, POLSTRACC, TACTS, WISE, START02 and most sondes and lidar records) but are not used in the article.
We have added two more figures in the appendix to highlight other datasets (SPURT, TACTS, Summit and South Pole ozonesondes and the Lauder lidar measurements).

We changed the text introducing Figure 7 to: Figure 7 displays various ozone datasets mapped using latitude versus altitude, a commonly utilized coordinate system. *These datasets were selected to showcase the variety of records included on OCTAV-UTLS: MLS and SAGEIII/ISS showcase satellite measurements with dense and sparse sampling, CARIBIC-2 illustrates aircraft measurements, while the Boulder and Table Mountain measurements represent the ozonesonde and lidar records, respectively.* As expected, ozone is low and well mixed in the troposphere, increasing in the lower stratosphere due to photochemical production.

2. Some figures, e.g. Figures 4 and 5, are rather small and it is difficult for the reader to check the coherence of the patterns for the various records (Figure 5).
We enlarged these figures to improve readability, we also enlarged Figure 7,8, A1 and A2.

In Figure 4, the distance from ground location for the satellite measurements needs to be explained.

We added in the caption: For satellites, distance from the ground location refers to the distance from the lowest available retrieval level.

3. Only 4 ground-based records are highlighted in Figures 7, 8, A1 and A2, while 14 ground-based records were processed for the article as mentioned in Table 1. The records' selection is not explained in the article. It would have been interesting to show more examples of ground-based climatological data in different coordinate systems. I also suggest to include a table providing the equivalent latitude range covered by the various ground-based records or the latitude range with respect to the subtropical jet.

As mentioned in the second response we added other figures to highlight more datasets.
We also added a figure showing the equivalent latitude and latitude wrt STJ coverage for each of the ozonesondes and lidars.

In the manuscript after the equivalent latitude Figure 7 discussion, we added the following sentences: Figure 8-left depicts a normalized count of the ozonesonde and lidar coverage in equivalent latitude. The vertical dashed lines display the geographical latitude for the ozonesonde launches or the lidar locations (noting that geographical and equivalent latitude have completely different meanings and implications). Any deviations observed between these lines and the maximum normalized count (i.e., 1) indicate that the respective datasets are situated in a region of significant dynamical variability, which leads to diabatic PV modification and thus a shift in equivalent latitude, such as the South Pole and Summit, Greenland ozonesondes, or the Table Mountain lidar measurements. As shown, the expanded space can cover from 11 to 30$^o$ (determined as the half width of the normalized counts) from the measurement location. Note that, in Figure 7, all other records show coverage improvements, with SAGEIII/ISS covering most of the globe, CARIBIC-2 extending its coverage throughout the northern hemisphere, and MLS covering the entire globe.

We also added, after the the latitude wrt STJ Figure 7 discussion: Note that when referring to latitude with respect to the STJ, all instruments show an expanded measurement space, which, is again most noticeable for the ozonesonde and lidar datasets. MLS, with its denser sampling, has full sampling of the region near the subtropical jets, as indicated by the concentric wind contours (black line) around the zero line. Figure 8-right also shows the extent of this coverage for the ozonesondes and lidars.

4. Differences in the climatological data displayed in Figures 7 and 8 need to be better described and explained. For instance, are the differences seen in MLS and SAGE III/ISS Theta/Eq latitude climatological data for equivalent latitudes > 60°S due to differences in sampling? In both figures, climatological data from CARIBIC-2 seem rather different. What is the reason for this and also for the differences in MLS and SAGEIII/ISS climatological temperature data?

For Figure 7, after the latitude altitude zonal mean discussion, we added:
The sampling of each record is evident in the zonal means presented here. MLS, with its dense sampling and near-global coverage (82∘S-82∘N), provides a comprehensive perspective of global

ozone fields, capturing the climatological view of the subtropical jet and subvortex jet (black contours, see Figure 6 for reference). However, MLS suffers from a lack of coverage below ~10 km and coarser vertical resolution compared to the other datasets used here. On the other hand, SAGEIII/ISS offers sampling in the upper troposphere with measurements down to 5 km, with better vertical resolution, but is limited to measurements between 60◦S and 60◦N. The distorted wind contours in comparison to MLS illustrate the greater impact of sampling biases than for MLS.

CARIBIC-2 measurements are restricted to flight levels, mostly situated at altitudes below 12 km, near the extratropical tropopause. These measurements, due to their high frequency observation rate, capture much finer details that are not resolved by satellite products. As such, it provides an exceptional perspective for studying tropopause-related processes in this region, but its coverage is limited elsewhere. Finally, ozone measurements obtained by ozonesondes and lidars provide superior vertical resolution, but are confined to their specific measurement locations. To enhance their visibility, we have replicated the ozonesonde and lidar measurements in the adjacent latitude bins in Figure 7-top.

The different sampling of each of these datasets may contribute to biases / artifacts and confound the interpretation of the results. To better understand this issue, OCTAV-UTLS aims to characterize the sampling biases of these datasets by comparing ozone reanalysis fields that are interpolated to the measurement times and locations (included in the dynamical diagnostics discussed herein) with the raw reanalysis fields (similar to the approach used in previous studies e.g., Toohey et al. (2013); Millán et al. (2016)).

After the tropopause mapping discussion, we added:

The impact on coverage of the tropopause mapping can be seen in particular in MLS and in CARIBIC-2. Although MLS shows very few measurements below the tropopause in the extratropics in the latitude/altitude view, mapping with respect to the tropopause allows separation of individual measurements that were taken above and below the tropopause. This shows that MLS does on occasion sample below the extratropical tropopause. Similarly, in the extratropics, CARIBIC-2 covers only a few kilometers above the tropopause in the latitude/altitude mapping, but the use of tropopause coordinates shows that it covers air masses that are high above the local tropopause (e.g., above deep folds) and thus samples deep stratospheric air.

In the Equivalent latitude mapping discussion, we added:

Figure 8-left depicts a normalized count of the ozonesonde and lidar coverage in equivalent latitude. The vertical dashed lines display the geographical latitude for the ozonesonde launches or the lidar locations (noting that geographical and equivalent latitude have completely different meanings and implications). Any deviations observed between these lines and the maximum normalized count (i.e., 1) indicate that the respective datasets are situated in a region of significant dynamical variability, which leads to diabatic PV modification and thus a shift in equivalent latitude, such as the South Pole and Summit, Greenland ozonesondes, or the Table Mountain lidar measurements. As shown, the expanded space can cover from 11 to 30º (determined as the half width of the normalized counts) from the measurement location. Note that, in Figure 7, all other records show coverage improvements, with SAGEIII/ISS covering most

of the globe, CARIBIC-2 extending its coverage throughout the northern hemisphere, and MLS covering the entire globe.

In the subtropical jet mapping discussion, we added:
Note that when referring to latitude with respect to the STJ, all instruments show an expanded measurement space, which, is again most noticeable for the ozonesonde and lidar datasets. MLS, with its denser sampling, has full sampling of the region near the subtropical jets, as indicated by the concentric wind contours (black line) around the zero line. Figure 8-right also shows the extent of this coverage for the ozonesondes and lidars.

For Figure 9 (old Figure 8) we added:
As was the case for ozone, to fully understand these differences, the sampling biases need to be characterized.

5. In Figures 7 and 8, red and orange lines are difficult to distinguish.
We changed the orange line to purple.

The black contours showing the wind speed are also different from one panel to the next. What are the contour values and what is the reason for the differences?
The black contours have been explained in the added text (see response above).
In the captions we added: The black contours show wind speed values of 30, 40, and 50 ms−1. Note that differences in their representation in comparison with MLS suggest sampling biases.

6. It would be useful to add some consideration in the conclusion about spurious trends in the reanalyses used to derive the diagnostic parameters and to which extent they could affect trend evaluation in alternative coordinate systems.
We added the following:
Thus, given these discontinuities, it is crucial to use the same reanalysis for all datasets throughout a given study. While using only one reanalysis can't completely eliminate the impact of these discontinuities, it can ensure that their effects are consistent across all datasets studied. That is, the timing of these discontinuities is well documented and it can be identified. If more than one reanalysis were used, the timing of discontinuities will be different, complicating the interpretation of such studies. That said, there is also value in repeating a study with meteorological fields from different reanalyses to study the sensitivity of the results to the uncertainties in the dynamical variables as recommended by S-RIP (Fujiwara et al., 2022).

We note that the conclusion already had the following sentence: Uncertainties arising from differences between reanalysis fields from different centers will also be examined by comparing the results when using different reanalyses as recommended by the SPARC Reanalysis Intercomparison Project.

7. A lot of self-citations. The authors could reduce the number of self-citations, in particular the Manney et al. citations (15 in the current version).
We reduced the number of Manney et al citations to 7. We cannot reduce it further since Manney's work laid the foundations for the work described here.

**Specific comments**

P2 L30-31: what is the reason for the trends in jet altitudes and velocities?
We changed the text to: For example, global tropopause altitudes and the frequency of double tropopauses increased substantially between 1981 and 2015 (e.g., Xian and Homeyer, 2019); subtropical jet strength and position trends show large regional and seasonal variations, but jet altitudes have typically increased (consistent with reports of tropopause altitude increases related to climate change) while their velocities have in general increased in winter and decreased in summer between 1980 and 2014 (e.g., Manney and Hegglin, 2018); the attribution of jet trends, particularly the polar (or "eddy-driven") jet, is an active research topic, with different studies arguing for either strengthening or weakening of the jets with climate change (e.g., Barnes and Screen, 2015; Francis, 2017; Manney and Hegglin, 2018).

P3 L6: to which measurement records did the satellite show a 10% bias in the zonal mean representation of ozone in the UTLS?
That study used sampled and raw model fields to evaluate the biases. We modified the sentence to: For example, Millán et al. (2016), using sampled and raw model fields, showed that even dense satellite sampling patterns may have up to 10% bias in the zonal mean representation of UTLS ozone.

P3 L17: equivalent latitude is not a tropopause related coordinate.
We have deleted Equivalent latitude

P4 L30: results from JOSIE experiments could also be cited here (e.g. Smit, H. G. J., et al. (2007), Assessment of the performance of ECC-ozonesondes under quasi-flight conditions in the environmental simulation chamber: Insights from the Juelich Ozone Sonde Intercomparison Experiment (JOSIE), J. Geophys. Res., 112, D19306, doi:10.1029/2006JD007308):
This citation was added

P4 L34: what is meant by homogenized?
We changed homogenized to gridded.

P5 L10: typo "typically".
Corrected

P5 L16: some studies are lacking in the list, e.g. Hubert et al., 2016, cited somewhere else in the article.
We have added the Hubert et al citation

Also, references are mostly related to both TMF lidar systems and could be diversified.
We also added the following citations through this section:
Godin-Beekmann et al., 2002 doi: 10.1039/B205880D
Raveta et al., 2007 doi: 10.1029/2006JD007724
Wing et al., 2020 doi: 10.5194/amt-13-5621-2020
Zeferos et al., 2018 doi: 10.5194/acp-18-6427-2018

P5 L25-35: The paragraph is difficult to read due to the number of acronyms. I suggest to refer to Table 2 for the campaign data.
We changed the paragraph to end on a list, listing the campaigns.

P6 L22: explain better the link between "oversampling" and the nominal vertical resolution of 1 km.
We added after oversampling: That is, measuring at vertical spacing finer than the instrument field-of-view of 3 km.

P7 L10: the whole NDACC ozone sondes records could be cited here.
We added: for example, the rest of the Network for the Detection of Atmospheric Composition Change (NDACC) ozonesonde records (e.g., Mazière et al., 2018), the ozonesondes included in the Southern hemisphere…

P7 L17: does the timeline of dynamical diagnostics correspond to the time range of the measurement records?
We added: Note that this timeline reflects the processing of the dynamical diagnostics and not the timeline of the measurement record, which in many cases extends to the current date.

P8 L30-32: the sentence is not clear: which other code could be used to compute the diagnostics?

We changed the text to:
When computing dynamical diagnostics for multiple datasets it is also important to use the same code to compute such diagnostics. For example, other algorithms exist to derive equivalent latitude (e.g., Hoor et al., 2004; Hegglin et al., 2006; Añel et al., 2013), jets information (e.g., Strong and Davis, 2008; Spensberger and Spengler, 2020, and references therein) or tropopause information (e.g., Cohen et al., 2021; Homeyer et al., 2021, and references therein). All of these algorithms involve several assumptions, which are different in different algorithms, and also numerous interpolations which may be done differently. Thus, significant differences could result simply from using a different code for different datasets.

P9 L14: potential temperature has not been introduced before log(theta) is mentioned here.
That paragraph now starts as:
The derived meteorological products are reanalysis fields, such as potential temperature ($\theta$), interpolated to the measurements' times and locations, as well as some derived meteorological products such as static stability (the gravitational resistance of the atmosphere to vertical displacements) and equivalent latitude (EqL).

P9 L22-23: the sentence is not clear, please reformulate.
We changed the sentence to: For example, Chapter 7 of the S-RIP report (Homeyer et al., 2021) illustrates that sampling biases can be a confounding factor even with dense sampling such as from MLS. Conducting sampling bias studies can aid in evaluating the comparability of datasets with very dissimilar sampling patterns, including those presented herein (see Figure 1). Moreover, sampling differences can result in significant biases in the inferred magnitude of ozone trends, as highlighted by (Millán et al., 2016).

P9 L28: MERRA-2 fields overestimate ozone and wind speed more than just slightly.
We removed the word *slightly*

P9 L30: for non-specialists, please explain tropopause inversions, especially with respect to double tropopauses.

We added the following: This figure also shows tropopause inversions (e.g., Birner et al., 2002, 2006). The tropopause inversion layer (TIL) is a shallow layer just above the tropopause, corresponding to a local maximum in static stability. These inversions often occur in the same regions and seasons as multiple tropopauses (e.g. Grise et al., 2010; Peevey et al., 2014; Schwartz et al., 2015). The representation of the TIL showcases the ability of modern reanalyses to capture the sharp gradients associated with such inversions (e.g., Gettelman and Wang, 2015; Kedzierski et al., 2016; Wargan and Coy, 2016). The ability to properly represent these inversion layers is important to define coordinates capable of capturing the trace gas gradients in the UTLS.

P10 L19: I thought that WMO tropopause is defined for a lapse rate below 2K km$^{-1}$.

The reviewer is correct for the first tropopause, for the second one (or more), the WMO requires a 3K km-1. To clarify this, we changed the sentence to: Note that to identify multiple thermal tropopauses we only require the temperature lapse rate to drop below 2 K km−1 as opposed to 3 K km−1 as defined by the WMO (WMO, 1957).

P10 L28: please provide pressure levels for the thermal and 4.5 PVU tropopauses.

We changed the paragraph to: On 21 December 2009, there was only one tropopause associated with the profile *(at 204 and 216 hPa for the 4.5 PVU and WMO tropopause, respectively),* while on 12 January 2010, two WMO tropopauses were identified by JETPAC *(at 201 and 98 hPa)*, coincident with strong gradients of static stability. On 12 January 2010, the locations of the primary WMO and 4.5 PVU tropopauses are almost the same.

P12 L5: please explain tropopause break.

We added: These criteria ensure that, in general, the identified subtropical jets are the ones close to the tropopause break (i.e., the abrupt drop in the height of the WMO tropopause while transitioning from the tropics to midlatitudes), consistent with primarily radiative driving, while the polar jets are consistent with primarily eddy driving.

P14 L12: it is not clear which fields the end of the sentence starting with "those fields" refers to.

We changed to: the reanalysis fields

P34 Table1: there are also two different lidar systems at OHP. Timespan for the stratospheric lidar is 1985 – present and that of the tropospheric is 1991 – present. The Hohenpeissenberg lidar records starts in 1987 and not 1978.

We updated the table accordingly.

---

## Author Comment (AC2)

We thank the reviewer for her/his comments. Below are our responses in blue.
Here is a summary of the major changes in the revised manuscript:

- We enlarged Figure 4, 5, 7, 8, A1 and A2 to improve readability.
- Figure A1 and A2 instead of showing Lauder lidar now show Mauna Loa so it can be compared against Hilo ozonesonde.
- We added two more figures in the appendix to highlight other datasets (SPURT, TACTS, Summit and South Pole ozonesondes and the Lauder lidar measurements).
- We change the title to: Multi-Parameter Dynamical Diagnostics for Upper Tropospheric and Lower Stratospheric Studies.
- We added a figure showing the range covered by the ozonesondes and lidars in EqL and latitude wrt STJ.
- We added discussions on the differences on sampling.
- We expanded the discussion of reanalysis discontinuities
- We emphasized the future OCTAV-UTLS studies with a figure showing some illustrative examples.

Reviewer 2 suggest that the manuscript is not suitable for this journal. The authors feel the material is appropriate for AMT, but if, after the revisions, the consensus among the reviewers and editor is that it is not, we will consider resubmitting (or transferring if that is possible) to ESSD. In any case, we thank the editor and reviewers in advance because we feel the paper is more complete after their suggestions.

**Review by Reviewer 2**
The paper is dedicated to several diagnostics for the UTLS studies, which are derived with the JETPAC software and MERRA2 reanalysis data. The authors introduce the SPARC OCTAV-UTLS project, which aims to reduce uncertainties in the UTLS trend estimates by accounting for dynamically induced variability. The authors state that "suitable coordinates should increase the homogeneity of the air masses analyzed together, thus reducing the uncertainty caused by spatio-temporal sampling biases in the quantification of UTLS composition trends".
The paper has a comprehensive introduction and describes well the objectives of the OCTAV-UTLS project. However, the scientific information content of the paper is low. The paper describes in detail the computing of dynamical diagnostics using the well-established JETPAC software, which is already described in several papers.
We agree on the fact that some of the JETPAC software is documented elsewhere, however, this manuscript offers a complete description of the current algorithm, that is, this manuscript offers a review and a compendium of all the changes / tweaks made since the original developments (i.e., Manney et al 2007 and Manney et al 2011). Originally, this algorithm was developed for satellite measurements but now it can process several type of datasets, including aircraft campaings, ozonesondes, and lidars. We believe there is value in having one manuscript describing this algorithm as opposed to having to read five or more papers to get the up-to-date picture. Our manuscript provides details that were not given in previous papers about the products that are produced at measurement locations, as opposed to the already well-established general JETPAC algorithm used for studies of the jets and tropopauses themselves in reanalysis data.

To emphasis these points we added the following sentences: Originally, these algorithms were developed to process satellite measurements, but they have since been adapted to accommodate a diverse range of

atmospheric measurements, including ozonesondes, lidars, and aircraft campaigns. Notably, this is the first time that all these records will be characterized consistently.

We also change 'For completeness, a full discussion is given here' to 'For completeness, a review of the previously published use of JETPAC products for characterizing composition measurements is given here along with an update describing the current capabilities (for example, processing high resolution datasets) and their applications. This review also describes the JETPAC algorithm as applied to characterizing measurement environments, as opposed to referring the reader to Manney et al. (2011) that describes a previous iteration of the algorithms for two satellite datasets, and Manney et al. (2014, 2017, 2021a), and Manney and Hegglin (2018) that use JETPAC products only on the reanalysis grids for dynamical studies. In other words, the previous publications do not give a full view of the current capabilities of JETPAC for characterizing composition measurements.'

Although examples of climatological ozone distributions in different coordinate systems are provided, there is no demonstration of advantages of using the alternative coordinates (references to published papers are not sufficient). In the current form, the paper does not meet the scope of AMT: "the development, intercomparison, and validation of measurement instruments and techniques of data processing and information retrieval for gases, aerosols, and clouds". The information related to atmospheric measurements, which is presented in the paper, is insufficient for AMT.

The authors (and reviewer 1) believe that the manuscript is in the scope of AMT: it shows intercomparisons and it shows techniques of data processing. We have expanded some discussions and highlighted additional datasets as detailed below.

From my point of view, there are two ways of improving the paper. The first one (preferable) is to add more illustrations and discussion on datasets in various coordinate systems and their agreement.

As requested by reviewer 1 we added two more figures in the appendix to highlight other datasets (SPURT, TACTS, Summit and South Pole ozonesondes and the Lauder lidar measurements).

We also added a figure showing the equivalent latitude and latitude wrt STJ coverage for each of the ozonesondes and lidars.

The reviewer is correct in pointing out that we could have shown some demonstrations of reduced error bars (i.e., due to correctly accounting for ozone gradients) or timeseries. However, those are the topics for OCTAV-UTLS studies that are in progress as described in the summary and future directions section.

To emphasize this, that section was rewritten as shown below. We also included illustrative examples (Figure 10 in the current manuscript), but we note that the actual analysis of those topics is left for future papers.

Future OCTAV-UTLS studies on ozone will:
– Investigate which dynamical coordinates better homogenize the mapped measurements, that is, which ones best segregate air in regions separated by geophysical transport barriers (for example, the tropopause) and thus provide the most comparable results within a given bin for non-coincident measurements. A paper on this topic is already on preparation. An illustrative example is shown on Figure 10-a for the WISE campaign using the relative standard deviation (RSTD, i.e., the standard deviation

divided by the mean) as metric to assess the variability in different coordinate systems. As expected, the dynamical coordinates display a smaller RSTD.

– Explore how sampling biases can confound the intercomparability of these records. These biases will be explored by comparing the ozone reanalysis fields interpolated to the measurement times and locations of the dynamical diagnostics discussed here to the raw reanalysis fields. Additionally, this study will explore which coordinates can, if any, reduce sampling biases. As an example, Figure 10-b shows the MLS sampling biases for January 2005.

– Analyze the impact of dynamical coordinates on quantification of long-term trends. The aim is to identify regions where dynamical coordinates can reduce the uncertainty associated with such trends. As an example, Figure 10-c shows MLS 20S-20N time series at 100 hPa as well as at the WMO tropopause. As shown, the time series at 100 hPa displays much larger amplitude variations that are related to the annual cycle of the 100 hPa pressure surface with respect to the tropopause, which results in sampling different fractions of stratospheric and tropospheric air in different seasons. Interannual differences in these variations are also likely related to changes in the relative altitudes of the 100 hPa isobaric surface and the tropopause.

Another way is shortening the paper and submitting it and the dataset of dynamical diagnostics to ESSD. We believe the manuscript is within the scope of AMT (since it provides intercomparisons and describes new techniques for data processing). By outlining the methods for data analysis and the complexities of intercomparing the diverse datasets (topics that are within the scope of AMT), this paper also provides necessary foundations for other OCTAV-UTLS studies that are in preparation and will need to cite it.

Reviewer 1 believed the manuscript to be publishable after some corrections. We have implemented those as detailed at the top of this document. We hope that these additions which provide more complete descriptions of the datasets and their analysis and relationships to physical phenomena, are sufficient to change reviewer 2's opinion on the manuscript.

OTHER COMMENTS
- The link to the dynamic diagnostics data does not work.
  The reviewer is correct, due to some JPL changes, we cannot longer host the dynamical diagnostics for the satellites under a JPL webpage. We are currently working to deliver the MLS dynamical diagnostics to a NASA disc. The dynamical diagnostics will be delivered to zenodo.

- It is mentioned that the collection of the datasets used in the paper is limited. With the selected subset of available data, it is probably possible to study some processes in the UTLS. However, for the ambitious objectives of OCTAV-UTLS, the dynamical diagnostics should be provided also for other available satellite and in-situ measurements.
  The reviewer is correct in suggesting that ideally the dynamical diagnostics should be process and made public for other satellites, ozonesondes, lidars, and aircraft campaigns to better attempt to fulfill the OCTAV-UTLS objectives. However, processing the dynamical diagnostic is time consuming and computationally expensive and currently there is no funding to process any more records. A MEASURES proposal was explicitly written to compute the dynamical diagnostics for many limb sounders, but the results of the panel review are still pending. Other avenues of research will be sought to process more datasets but, in the meantime, no more records can be process.

Alternatively, JETPAC can be made as a free software.

Unfortunately, at this time, the JETPAC software cannot be made publicly available due to JPL regulations, but we will explore pathways to make it public in the future.

- P.7 " When computing dynamical diagnostics as discussed in this section it is important to use the same reanalysis fields for all the datasets to be used in a given study." However, possible discontinuities in reanalyses data, which are caused by changes in assimilated datasets (your discussion on Page 8), will also affect the dataset of dynamical diagnostics. When the same reanalysis is used for all datasets, these discontinuities will appear as an artificial drift in the dynamical parameters and thus in evaluated trends (if these dynamical parameters are used for data transformation). When using different reanalyses, one may hope that the timing of discontinuities will be different, and thus the overall evaluation of the trends using multiple datasets will have a reduced reanalyses-related drift. These issues should be discussed/mentioned in the paper.

  The reviewer is right, that discontinuities could affect the interpretation of data. Here, first, we wanted to make sure that the different observational characteristics of the different data sets are related to the same reanalysis fields to ensure a consistency at this point. Those issues were discussed in the paper at the end of the same paragraph discussed in the reviewer comment. However, to emphasis the discussion we broke up the paragraph and added: Thus, given these discontinuities, it is crucial to use the same reanalysis for all datasets throughout a given study. While using only one reanalysis can't completely eliminate the impact of these discontinuities, it can ensure that their effects are consistent across all datasets studied. That is, the timing of these discontinuities is well documented and it can be identified. If more than one reanalysis were used, the timing of discontinuities will be different, complicating the interpretation of such studies. That said, there is also value in repeating a study with meteorological fields from different reanalyses to study the sensitivity of the results to the uncertainties in the dynamical variables as recommended by S-RIP (Fujiwara et al., 2022).

- The influence of sampling patterns should be discussed in more detail. In particular, it should be mentioned /discussed also in the text related to Figure 5 and Figure 7. The distributions for CARIBIC-2 data are different from those for other datasets.

  We added the following after the latitude altitude zonal mean discussion for Figure 7:

  The sampling of each record is evident in the zonal means presented here. MLS, with its dense sampling and near-global coverage (82◦S-82◦N), provides a comprehensive perspective of global ozone fields, capturing the climatological view of the subtropical jet and subvortex jet (black contours, see Figure 6 for reference). However, MLS suffers from a lack of coverage below ∼10 km and coarser vertical resolution compared to the other datasets used here. On the other hand, SAGEIII/ISS offers sampling in the upper troposphere with measurements down to 5 km, with better vertical resolution, but is limited to measurements between 60◦S and 60◦N. The distorted wind contours in comparison to MLS illustrate the greater impact of sampling biases than for MLS.

  CARIBIC-2 measurements are restricted to flight levels, mostly situated at altitudes below 12 km, near the extratropical tropopause. These measurements, due to their high frequency observation rate, capture much finer details that are not resolved by satellite products. As such, it provides an exceptional perspective for studying tropopause-related processes in this region, but its coverage is limited elsewhere. Finally, ozone measurements obtained by ozonesondes and lidars provide

superior vertical resolution, but are confined to their specific measurement locations. To enhance their visibility, we have replicated the ozonesonde and lidar measurements in the adjacent latitude bins in Figure 7-top.

The different sampling of each of these datasets may contribute to biases / artifacts and confound the interpretation of the results. To better understand this issue, OCTAV-UTLS aims to characterize the sampling biases of these datasets by comparing ozone reanalysis fields that are interpolated to the measurement times and locations (included in the dynamical diagnostics discussed herein) with the raw reanalysis fields (similar to the approach used in previous studies e.g., Toohey et al. (2013); Millán et al. (2016)).

After the tropopause mapping discussion, we added:

The impact on coverage of the tropopause mapping can be seen in particular in MLS and in CARIBIC-2. Although MLS shows very few measurements below the tropopause in the extratropics in the latitude/altitude view, mapping with respect to the tropopause allows separation of individual measurements that were taken above and below the tropopause. This shows that MLS does on occasion sample below the extratropical tropopause. Similarly, in the extratropics, CARIBIC-2 covers only a few kilometers above the tropopause in the latitude/altitude mapping, but the use of tropopause coordinates shows that it covers air masses that are high above the local tropopause (e.g., above deep folds) and thus samples deep stratospheric air.

In the Equivalent latitude mapping discussion, we added:

Figure 8-left depicts a normalized count of the ozonesonde and lidar coverage in equivalent latitude. The vertical dashed lines display the geographical latitude for the ozonesonde launches or the lidar locations (noting that geographical and equivalent latitude have completely different meanings and implications). Any deviations observed between these lines and the maximum normalized count (i.e., 1) indicate that the respective datasets are situated in a region of significant dynamical variability, which leads to diabatic PV modification and thus a shift in equivalent latitude, such as the South Pole and Summit, Greenland ozonesondes, or the Table Mountain lidar measurements. As shown, the expanded space can cover from 11 to  30$^{o}$ (determined as the half width of the normalized counts) from the measurement location. Note that, in Figure 7, all other records show coverage improvements, with SAGEIII/ISS covering most of the globe, CARIBIC-2 extending its coverage throughout the northern hemisphere, and MLS covering the entire globe.

In the subtropical jet mapping discussion, we added:
Note that when referring to latitude with respect to the STJ, all instruments show an expanded measurement space, which, is again most noticeable for the ozonesonde and lidar datasets. MLS, with its denser sampling, has full sampling of the region near the subtropical jets, as indicated by the concentric wind contours (black line) around the zero line. Figure 8-right also shows the extent of this coverage for the ozonesondes and lidars.